# *Cambrian*-1: A Fully Open, *Vision-Centric* Exploration of Multimodal LLMs

**Shengbang Tong**[*]   **Ellis Brown**[*]   **Penghao Wu**[*]
**Sanghyun Woo**   **Manoj Middepogu**   **Sai Charitha Akula**   **Jihan Yang**
**Shusheng Yang**   **Adithya Iyer**   **Xichen Pan**   **Austin Wang**
**Rob Fergus**   **Yann LeCun**   **Saining Xie**[†]
New York University

## Abstract

We introduce Cambrian-1, a family of multimodal LLMs (MLLMs) designed with a vision-centric approach. While stronger language models can enhance multimodal capabilities, the design choices for vision components are often insufficiently explored and disconnected from visual representation learning research. This gap hinders accurate sensory grounding in real-world scenarios. Our study uses LLMs and visual instruction tuning as an interface to evaluate various visual representations, offering new insights into different models and architectures—self-supervised, strongly supervised, or combinations thereof—based on experiments with over 20 vision encoders. We critically examine existing MLLM benchmarks, addressing the difficulties involved in consolidating and interpreting results from various tasks, and introduce a new vision-centric benchmark, CV-Bench. To further improve visual grounding, we propose the Spatial Vision Aggregator (SVA), a dynamic and spatially-aware connector that integrates high-resolution vision features with LLMs while reducing the number of tokens. Additionally, we discuss the curation of high-quality visual instruction-tuning data from publicly available sources, emphasizing the importance of data source balancing and distribution ratio. Collectively, Cambrian-1 not only achieves state-of-the-art performance but also serves as a comprehensive, open cookbook for instruction-tuned MLLMs. We provide model weights, code, supporting tools, datasets, and detailed instruction-tuning and evaluation recipes. We hope our release will inspire and accelerate advancements in multimodal systems and visual representation learning.

**Project page:** https://cambrian-mllm.github.io/

## 1  Introduction

There is a long-standing debate in philosophy about whether understanding and meaning in language require sensory grounding. Aristotle's emphasis on acquiring knowledge through sensory experience and empirical observation was central to his ancient Peripatetic school and remains influential to this day [8]; Aquinas famously formalized these ideas in the 13th century with the Peripatetic axiom: "*Nihil est in intellectu quod non sit prius in sensu*" (Nothing is in the intellect that was not first in the senses) [7]. Though many philosophers disagree [23], it is evident that having robust and highly capable sensory grounding is at least beneficial. Consider the *Cambrian explosion*, during which the emergence of vision is believed [105] to have been crucial for early animals to not only find food and avoid predators but also to evolve and improve. In fact, most human knowledge (and nearly

---

[*]Project Lead
[†]Corresponding Author

38th Conference on Neural Information Processing Systems (NeurIPS 2024).

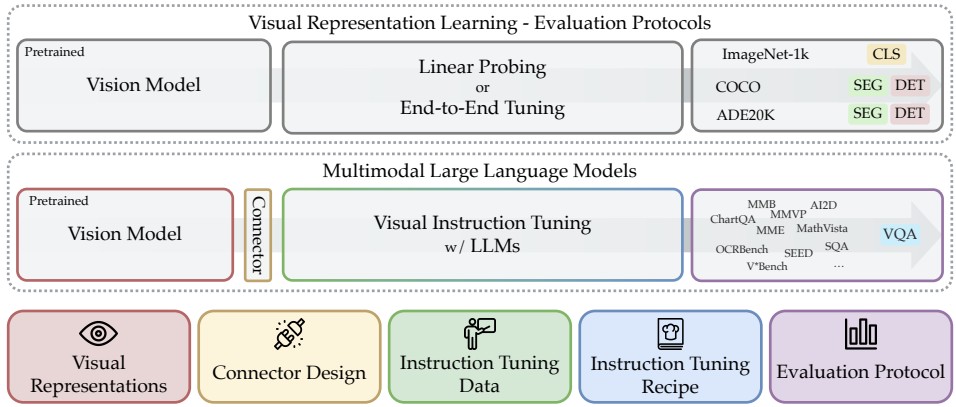

Figure 1: We draw parallels between traditional protocols and the use of MLLMs for evaluating visual representations. MLLMs employ visual question answering to address a diverse array of real-world perception tasks. The bottom section highlights the five key pillars studied in Cambrian-1.

all animal knowledge) is acquired through sensory experiences like sight, hearing, touch, taste, and smell, through interactions with the physical world [107]. These sensory experiences are fundamental to understanding the world around us and are crucial for real-world actions and decision-making.

Beyond philosophical debates, recent advances in multimodal large language models (MLLMs) have brought the topic of *visual representation learning vs. language understanding* into practical focus. Language models have shown strong scaling behaviors [55], and recent advancements in multimodal learning are largely driven by the development of better, larger LLMs [81]. On the other hand, the design choices for vision components are often insufficiently explored and *disconnected* from visual representation learning research. For instance, many pioneering frameworks such as LLaVA [82] use vision transformer-based CLIP models [109, 145], which are strongly supervised by language[‡], as the vision feature extractor. While other visual representations, such as self-supervised DINO [103], are being explored [126], there is a lack of comprehensive and systematic study in this domain. This gap exists primarily because such studies are challenging: MLLMs involve a complex training and evaluation pipeline with numerous design decisions to consider. In this work, we aim to bridge the gap by exploring MLLMs from a vision-centric perspective. More specifically, we use MLLM instruction tuning as an evaluation protocol for various visual representations (illustrated in Fig. 1).

Our motivation for this study also stems from two potential concerns of the current multimodal learning research: 1) relying too heavily too early on language can act as a shortcut [47, 144], compensating for the deficiencies in learning effective visual representations, and 2) existing benchmarks may not provide adequate guidance for real-world scenarios—where visual grounding is crucial for robust multimodal understanding. These concerns are not unfounded, as researchers have started to notice that visual grounding is becoming a bottleneck for applying MLLMs in some challenging real-world applications, despite significant progress in improving general capabilities [41, 126, 136].

From another perspective, traditional evaluation protocols for visual representation learning (e.g., *linear probing* and *end-to-end fine-tuning* on datasets like ImageNet-1K [113], COCO [79], and ADE20K [154]) are becoming saturated and do not reflect the diverse perception challenges found in real-world distributions. On the other hand, using language in the form of visual question answering (VQA) offers a flexible and robust evaluation protocol. Our study aims to explore this new protocol design, setting it up to gain insights that will guide the development of better visual representations in the future. Furthermore, to better evaluate visual representations in this integrated setting, we develop a vision-centric MLLM benchmark, CV-Bench, by transforming traditional vision benchmarks into VQA format (Section 2.2).

Cambrian-1 is structured around five key pillars, each offering important insights into the design space of MLLMs:

- **Visual Representations**: We explore various vision encoders and their combinations. §2.4

---

[‡]We emphasize that CLIP training should be considered as *strongly supervised*, as language provides significantly richer supervision than class labels.

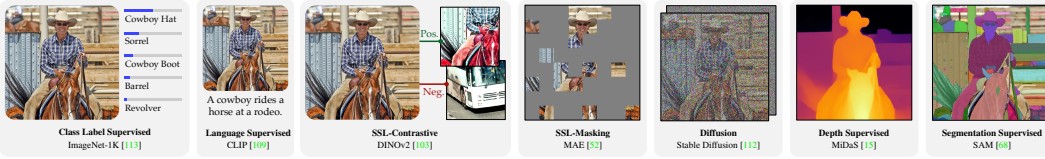

Figure 2: Examples of various vision models, objectives, and architectures studied. Image from [48].

- **Connector Design**: We design a new dynamic and spatially-aware connector that integrates vision features with LLMs while reducing the number of tokens. §3
- **Instruction Tuning Data**: We curate high-quality visual instruction-tuning data from public sources, emphasizing the importance of distribution balancing. §4
- **Instruction Tuning Recipes**: We discuss instruction tuning strategies and practices. §2.3
- **Benchmarking**: We analyze existing MLLM benchmarks, cluster them into 4 intuitive groups, and introduce a new vision-centric benchmark "CV-Bench". §2.1, §2.2

We defer a detailed review of the fundamental components and methodologies that underpin MLLM research to Appendix B

## 2 Evaluating Visual Representations through MLLMs

Current MLLMs predominantly rely on CLIP [109] as the visual encoder due to its pre-alignment with language and ease of adaptation to the LLM token space. However, strong language priors can be a double-edged sword—they compensate for deficiencies in learning effective visual representations [126] and diminish insights gained from extensive visual representation learning research. In this section, we systematically evaluate how various visual encoder choices (see Fig. 2) impact the multimodal capabilities of MLLMs. We also advocate for using MLLM evaluation as a robust framework for assessing visual representation methods, moving beyond traditional protocols like linear probing and end-to-end fine-tuning to more faithfully reflect the diverse perception challenges in real-world scenarios and to better guide the development of improved visual representations.

### 2.1 Analyzing the Benchmarks

To effectively evaluate visual representations and MLLMs, we first need to select benchmarks that accurately assess the *multimodal* capabilities of these models. We use a suite of commonly used benchmarks [24, 45, 54, 57, 83, 84, 91, 92, 96, 97, 120, 126, 137, 143], which is the intersection of those used in recent MLLM research [75, 77, 137]. To help interpret our results, we begin by analyzing the benchmarks themselves. Here, we train MLLMs with 23 different vision backbones (see Table 6) from a variety of model families (see Fig. 2) using a 2-stage instruction tuning process initially proposed in [82]: first training connector on 1.2M adapter data from ShareGPT-4V [27] followed by fine-tuning both the connector and LLM on 737K instruction tuning data (see more details in Appendices G.5 and H). Full benchmark results in Table 9.

**Who's answering the question: the LLM or MLLM?** Determining whether a benchmark *truly* needs visual input to be solved has been a persistent challenge in vision-language research [2, 26, 50, 94]. In this study, we compare the performance of MLLMs with and without visual input[§], and also calculate the expected score via randomly guessing. These three conditions are visualized in Fig. 3-left, with benchmarks sorted by the difference between the average score with vision enabled and disabled. SQA-I[¶], MMMU, MathVista, and AI2D display less than a 5% gap between vision enabled and disabled, suggesting that these benchmarks may not significantly depend on visual input and rather heavily rely on the base LLM. TextVQA and GQA both demonstrate a nearly 40% positive gap between random guessing and vision-disabled scores, implying a strong language bias in these benchmarks. On the other hand, the vision-disabled performance on benchmarks like MMVP is notably worse than random guessing, suggesting that strong visual grounding is particularly crucial.

**Clustering the Benchmarks** To better understand the different aspects of MLLM performance, we analyze the correlations between the performance of our 23 MLLMs on each benchmark. A

---

[§]We note that our instruction-tuning data includes text-only data, so text-only questions are not OOD.

[¶]The subset of SQA [91] with images.

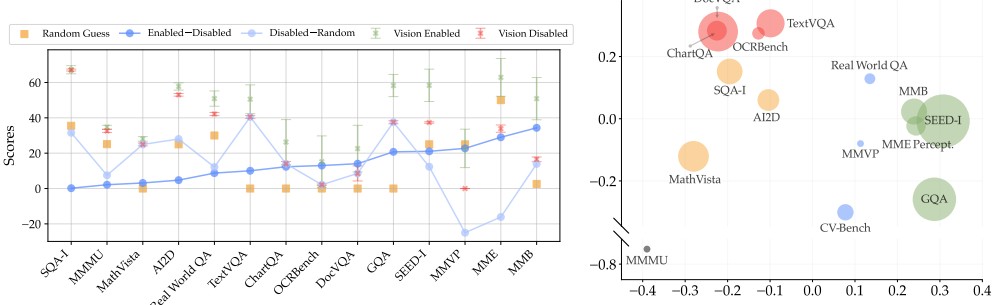

Figure 3: **Left:** Performance comparison of MLLMs with visual input enabled and disabled across various benchmarks. Benchmarks are sorted by the difference between the average score with vision enabled and disabled. **Right:** Principal component analysis displaying clusters of benchmarks based on performance metrics, with bubble size corresponding to benchmark size. We label the clusters as "General" in green, "Knowledge" in yellow, "Chart & OCR" in red, and "Vision-Centric" in blue.

confusion matrix (Fig. 10) reveals that certain benchmarks, such as MMMU, are largely uncorrelated with the others. We perform principal component analysis on the benchmark scores and observe the formation of clusters corresponding to "General," "Knowledge," "Chart & OCR," and "Vision-Centric" categories (Fig. 3-right). We assign MMMU to the knowledge category based on the types of questions it includes (see Appendix D). We also find that existing vision-centric benchmarks [126, 137] are of insufficient size (see Fig. 3-right), challenging the robustness of evaluating such capabilities. These benchmarks do not cover crucial visual elements such as depth and spatial awareness.

> **Finding 1:** Most benchmarks do not properly measure vision-centric capabilities, and the ones that do have very few samples.

## 2.2 Cambrian Vision-Centric Benchmark (CV-Bench)

To address the limitations of existing vision-centric benchmarks, we introduce the Cambrian Vision-Centric Benchmark (CV-Bench). With **2638 manually-inspected examples**, CV-Bench provides significantly more examples than other vision-centric MLLM benchmarks—$3.5\times$ more than Real-WorldQA [137] and $8.8\times$ more than MMVP [126]. By repurposing standard vision benchmarks [18, 79, 154][||], we can assess models at classic vision tasks within a multimodal context. Leveraging the rich ground truth annotations from the benchmarks, we formulate natural language questions that probe the fundamental 2D and 3D understanding of the models.

As visualized in Fig. 11, CV-Bench evaluates 2D understanding via spatial relationships & object counting, and 3D understanding via depth order & relative distance. We refer details to Appendix E.

> **Finding 2:** Existing vision benchmarks can be effectively repurposed into VQA questions, enabling the assessment of vision-centric MLLM capabilities.

## 2.3 Instruction Tuning Recipes

MLLMs start with pre-trained LLM and vision backbones, connecting these modules with a connector such as a projector (MLP). The original LLaVA [80, 82] proposes a 2-stage frozen training process: first, pre-training a connector between frozen LLM and vision backbones using adapter data, and then fine-tuning both the connector and LLM with instruction tuning data while leaving the vision encoder frozen. Various studies [27, 63, 81, 98] have drawn different conclusions regarding the optimal training methodology for MLLMs. Here, we revisit this topic with extensive experiments.

For our experiments, we tune a set of MLLMs using Vicuna-1.5-7B as the LLM backbone and each of our 23 vision models (Table 6) as the visual encoder. We use a 737K instruction tuning data mix for all experiments here (see Appendix H). All hyperparameters are matched across each experimental setting—highlighting the impact of different tuning strategies with each visual encoder. All experimental settings and results are tabulated in Appendix F.2.

---

[||]Omni3D assets are sourced from [3, 13, 20, 46, 111, 121].

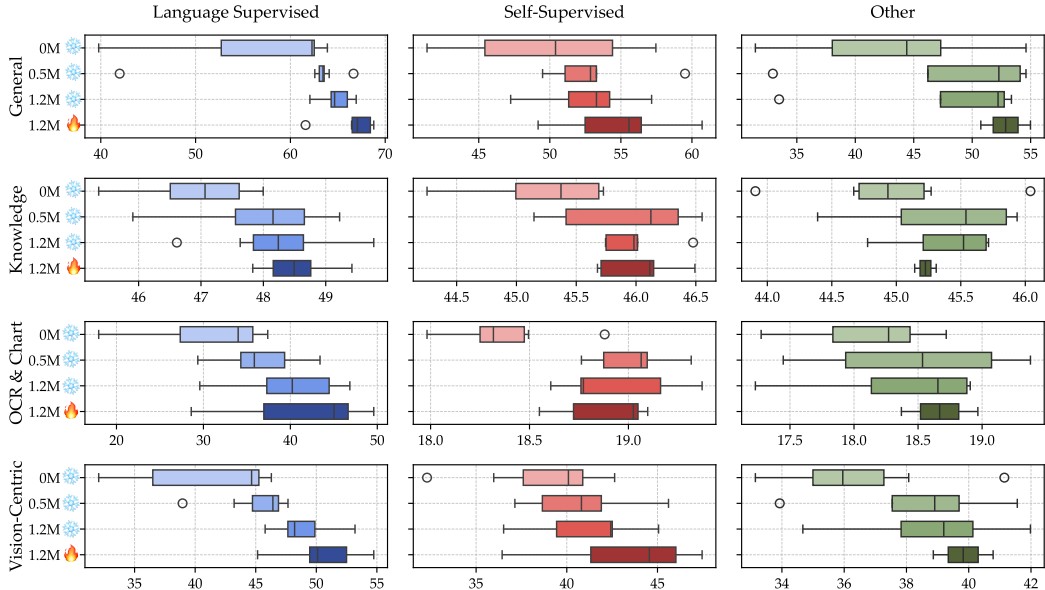

Figure 4: **Effect of Training Recipe on Model Performance**. Boxplots display the distribution of benchmark scores across benchmark categories for different training recipes and types of visual encoders. The four training recipes include freezing the visual encoder with various amounts of adapter data (0M❄, 0.5M❄, 1.2M❄) as well as unfreezing it with 1.2M🔥 adapter data. **Amount of Adapter Data**: All model types show increased performance on general and vision-centric benchmarks; knowledge benchmarks show mixed results; OCR & chart benchmarks benefit from more data for language-supervised models. **Unfreezing**: Unfreezing the visual encoder with 1.2M🔥 adapter data generally benefits all categories.

**One Stage vs Two Stage Training**   Recent work [63] advocates for skipping connector pre-training, claiming this "*reduces compute cost without harming downstream performance*." To explore whether this claim holds—especially when using non-language-supervised visual encoders—we conduct experiments using 0, 0.5M, and 1.2M adapter data. Following LLaVA's recipe [82], we tune only the connector on the adapter data during this first phase, before unfreezing the LLM and connector during instruction tuning on the 737K mix. Fig. 4 shows that pre-training the connector first enhances model performance and that more adapter data further improves performance across all domains. Thus, we subsequently adopt 2-stage training with 1.2M adapter data as our standard setup.

> *Finding 3:*   Two-stage training is beneficial; more adapter data further improves results.

**Freeze vs Unfreeze Vision Encoder**   There are also mixed practices in freezing [63, 80, 82] or unfreezing [44, 81] vision backbones during fine-tuning. Some argue that unfreezing the vision backbone significantly degrades performance [63]. Our experiments demonstrate that unfreezing benefits performance across all benchmarks except for a marginal change in knowledge benchmarks (Fig. 4). We suspect this is due to the composition of the 737K instruction tuning data and the LLM-heavy focus of these benchmarks (see Section 2.1). We note that unfreezing the vision backbone introduces additional computational overhead, which prohibits testing on some larger vision models under current sharding strategies (see more details in Appendix H).

> *Finding 4:*   Unfreezing the vision encoder is widely beneficial. Language-supervised models always benefit; SSL models particularly benefit on vision-centric benchmarks.

## 2.4   MLLMs as a Visual Representation Evaluator

As discussed in earlier sections, MLLMs provide a new interface to explore aspects of vision models beyond traditional benchmarks like ImageNet-1k linear probing. We study the 2-stage instruction tuning setting using 1.2M adapter data, 737K fine-tuning data, and frozen visual encoders to allow comparison of the widest range of models.

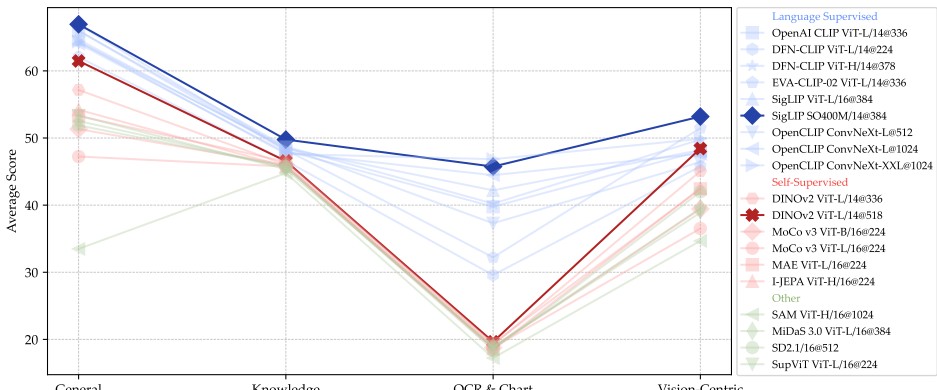

Figure 5: **Evaluating Visual Representations with MLLMs** While language-supervised models outperform self-supervised or other models, a well-trained self-supervised model like DINOv2 can also achieve competitive performance on vision-centric tasks.

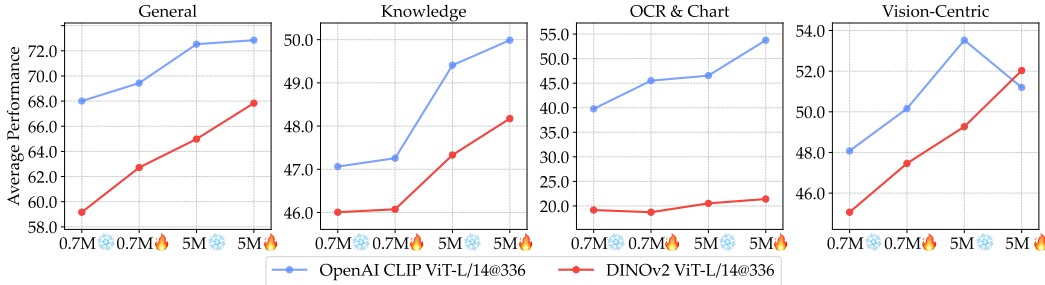

Figure 6: **Continued Fine-Tuning Narrows the Gap Between CLIP and DINOv2**. Performance is compared with 0.7M and 5M instruction tuning data in both frozen (❄) and unfrozen (🔥) settings. DINOv2 shows significant performance improvement with increased data and unfreezing—surpassing the 0.7M❄ CLIP model in several benchmarks and narrowing the gap to the 5M🔥 model in knowledge and vision-centric tasks.

We evaluate on benchmarks detailed in Section 2.1, calculating the average performance** for each category and visualize the results in Fig. 5 (full results in Appendix F). Our findings highlight the advantages of language-supervised models over non-CLIP models across all benchmark categories, with significantly better performance on chart and OCR-related benchmarks. We hypothesize that this is due to CLIP's *training data*, such as LAION [115], containing abundant OCR and text-heavy data, whereas SSL and other vision models primarily train on natural images with significantly less text content. It is also noteworthy that language-supervised models are typically trained with a very large pool of data, ranging from 400 million [109] to 10 billion [28] samples, whereas the largest vision self-supervised training dataset, like DINOv2, consists of only *142 million samples* [103].

Additionally, we observe that higher-resolution models particularly enhance performance on chart and vision-centric benchmarks while remaining neutral on general VQA and knowledge-based VQAs. While the majority of the backbones we examine are ViT-based [39], **ConvNet-based architectures** (such as OpenCLIP ConvNeXt [86]) are inherently well-suited for high-resolution image processing [130] and can produce superior results on OCR & Chart and Vision-Centric benchmarks. In vision-centric benchmarks, the gap between language-supervised and other types of vision models is smaller, with a well-trained self-supervised DINOv2 model even outperforming some language-supervised models.

> *Finding 5:* High-res encoders greatly enhance performance on chart & vision-centric benchmarks, and ConvNet-based architectures are inherently well-suited for such tasks.

**Narrowing the gap between Language- and Self-Supervised models** Above, we observe that DINOv2 stands midway between SSL models and language-supervised models on general and knowledge benchmarks, even outperforming some language-supervised models on vision-centric benchmarks. Here, we study whether the continued finetuning of an MLLM based on a SSL model

---

**Before averaging, we divide the MME Perception score by 20 to have the same scale as other benchmarks.

can achieve performance similar to that of a language-supervised model. Specifically, we scale up the instruction tuning data from 737K to 5M (see more details in Appendix G.5), and instruction tune MLLMs with DINOv2 ViT-L/14@336 and OpenAI CLIP ViT-L/14@336 encoders in both frozen and unfrozen settings. In Fig. 6, we observe that by unfreezing the vision backbone, the DINOv2-based MLLM fine-tuned with 5M data surpasses the MLLM trained with a CLIP model on 0.7M data. Additionally, the gap between DINOv2 and the CLIP models is reduced under the 5M setting.

> ***Finding 6:*** Language supervision offers strong advantages, but the performance gap can be narrowed with SSL methods given enough data and proper tuning.

### 2.5 Combining Multiple Vision Encoders

As observed in Fig. 5, different vision encoders excel in different aspects of MLLM performance. In this study, we explore the potential of combining multiple vision encoders to leverage their distinctive representations, aiming to build a more capable MLLM. Given that different vision encoders use varying architectures and image resolutions, we interpolate to a fixed number of visual tokens (576) in this subsection (see details in Appendix F.3). We then concatenate these tokens along the feature dimension, following a method similar to A-MoF proposed in [126].

Our study indicates that adding a non-language-supervised model (DINOv2) can improve benchmark performance, especially in vision-centric tasks. Notably, even OCR benchmarks benefit from incorporating DINOv2. This highlights the importance of self-supervised learning models in complementing language-supervised models to achieve robust multimodal understanding. Detailed results and configurations are available in Appendix F.3.

However, this naive strategy has two limitations: 1) it employs interpolation, which can lead to information loss, especially with vision encoders with high-resolution feature maps, and 2) it treats each model equally via simple concatenation. Therefore, we seek a more effective strategy that can more flexibly leverage model combinations with less information loss.

> ***Finding 7:*** Combining multiple vision encoders, including SSL models, can enhance MLLM performance across various benchmarks, particularly in vision-centric tasks.

## 3  Spatial Vision Aggregator (SVA): A New Connector Design

To effectively aggregate features from multiple vision encoders and prevent the information loss introduced by interpolation, we use a set of learnable latent queries that interact with multiple vision features via cross-attention layers [37]. In particular, our approach incorporates two new vision-centric design principles:

1. We introduce spatial inductive bias by explicitly defining the aggregation space for each token in the query.

2. We aggregate vision features multiple times across the LLM layers, enabling the model to repeatedly access and integrate necessary visual information.

To facilitate information aggregation via cross-attention, we create a $C$-dimension learnable latent token $\mathbf{x} \in \mathbb{R}^C$ that is repeated $L \times L$ times to form a 2D grid, serving as the query $\mathbf{X} \in \mathbb{R}^{L^2 \times C}$. The set of visual features $\mathbf{F}$ from $N$ vision encoders serve as the context (i.e., key and value). We ensure the output resolution of every vision encoder is a multiple of $L$. Formally, the feature map of the $k$-th vision encoder ($\mathbf{F}_k$) has a resolution of $m_k L \times m_k L \times C$, where $m_k$ is a positive integer multiplier, and $L$ is the height/width of the learnable 2D grid with hidden dimension $C$.

**Spatial inductive bias**  To maintain the spatial structure during cross-attention, we align each token in the query with a specific sub-region of the feature maps in all vision encoders. Formally, a token at row $i$ and column $j$ in the query $\mathbf{x}_{i,j}$ corresponds to the sub-region

$$\mathbf{F}_k[m_k \cdot i : m_k \cdot (i+1), m_k \cdot j : m_k \cdot (j+1)] \in \mathbb{R}^{m_k^2 \times C}$$

of the $k$-th vision feature map. As a result, a token $\mathbf{x}_{i,j}$ aggregates a total of $\sum_k m_k^2$ features from $N$ vision encoders through cross-attention (see Fig. 7-left).

Specifically, the updated query vector $\mathbf{q}^*_{\mathbf{i,j}} \in \mathbb{R}^{1 \times C}$ at position $(i, j)$ is computed as

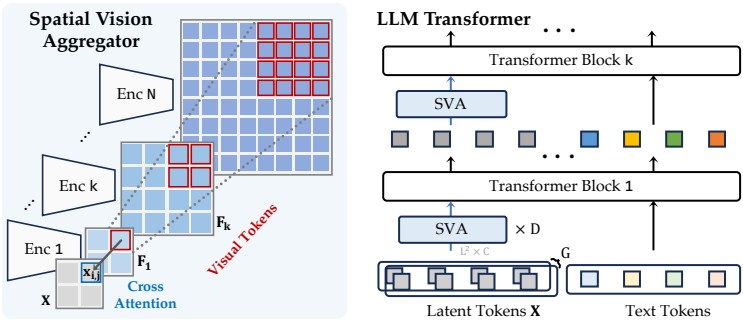

Figure 7: **Spatial Vision Aggregator (SVA).** We propose SVA, a dynamic and spatially-aware connector that integrates multiple vision features with LLMs while reducing the number of tokens.

$$\mathbf{q}^*_{i,j} = \mathrm{softmax}\left(\frac{\mathbf{q}_{i,j} \cdot [\mathbf{k}_{i,j,1}, \mathbf{k}_{i,j,2}, \ldots, \mathbf{k}_{i,j,N}]^\top}{\sqrt{C}}\right)[\mathbf{v}_{i,j,1}, \mathbf{v}_{i,j,2}, \ldots, \mathbf{v}_{i,j,N}], \tag{1}$$

where

$$\mathbf{q}_{i,j} = \mathbf{W}^Q \mathbf{x}_{i,j} \in \mathbb{R}^{1 \times C},$$
$$\mathbf{k}_{i,j,k} = \mathbf{W}^K_k \mathbf{F}_k[m_k \cdot i : m_k \cdot (i+1),\ m_k \cdot j : m_k \cdot (j+1)] \in \mathbb{R}^{m_k^2 \times C},$$
$$\mathbf{v}_{i,j,k} = \mathbf{W}^V_k \mathbf{F}_k[m_k \cdot i : m_k \cdot (i+1),\ m_k \cdot j : m_k \cdot (j+1)] \in \mathbb{R}^{m_k^2 \times C}.$$

Here, $\mathbf{q}_{i,j}$ is the query vector at position $(i, j)$, calculated using the query projection matrix $\mathbf{W}^Q \in \mathbb{R}^{C \times C}$. The key vectors $\mathbf{k}_{i,j,k}$ and value vectors $\mathbf{v}_{i,j,k}$ are computed for each vision encoder $k$ using their respective key and value projection matrices $\mathbf{W}^K_k \in \mathbb{R}^{C \times C}$ and $\mathbf{W}^V_k \in \mathbb{R}^{C \times C}$. Since $\sum_k m_k^2$ features are aggregated into a single token, we effectively reduce the number of tokens.

**Multi-layer vision aggregation** Although our proposal effectively aggregates features from multiple vision encoders, there is still potential information loss with high-resolution input (large $m_k$) or multiple vision encoders (large $N$). Here, a single token would have to handle a larger amount of context information during aggregation. To prevent this, we allow cross-attention to occur multiple times by inserting our proposal throughout the LLM layers—allowing consistent access to the uncompressed visual information (see Fig. 7-right).

**Hyperparameters** To flexibly modulate capacity, we introduce two hyperparameters $D$ and $G$, which indicate the number of cross-attention layers and distinct groups of learnable queries used between the vision models and the LLM, respectively. $D$ and $G$ are always set to 1 for cross-attention layers within LLM layers. We provide ablation studies on the selection of $D$ and $G$ in Appendix H.

| Connector | General | Knowledge | OCR & Chart | Vision-Centric |
|---|---|---|---|---|
| Concat. [126] | 67.2 | 48.9 | 50.1 | 52.6 |
| Resampler [58] | 63.1 | 46.5 | 27.1 | 42.6 |
| SVA-no-multi-agg | 68.0 | 49.5 | 55.2 | 52.6 |
| **SVA** | **68.5** | **49.7** | **55.5** | **53.2** |

Table 1: **Comparison between our SVA and other aggregation approaches.** The SVA module consistently outperforms other baselines and excels in aggregating high-resolution vision information.

We demonstrate the efficacy of SVA module using the best vision model combination results from the previous section and a Vicuna-1.5-7B base LLM. Specifically, we employ a combination of four vision encoders: OpenAI CLIP ViT-L/14@336, SigLIP ViT-SO400M/14@384, OpenCLIP ConvNeXt-XXL@1024, and DINOv2 ViT-L/14@518. We compare our method with two strong baselines: 1) concatenation-based [126] and 2) Re-sampler [11, 72]. Here, we include two variants of our SVA module. The standard one, "**SVA**", uses $D = 3$, $G = 1$, and inserts cross-attention blocks inside the LLM with a layer stride of 3. To isolate the advantages of spatial inductive biases, we include another SVA variant, "SVA-no-multi-agg", that does not add cross-attention blocks inside the LLM and sets $D = 3$ and $G = 3$. Table 1 shows that SVA outperforms both baselines, with a significant improvement in the OCR & chart category. In contrast, the Resampler—which lacks spatial inductive biases—struggles to condense concatenated tokens from various vision towers into

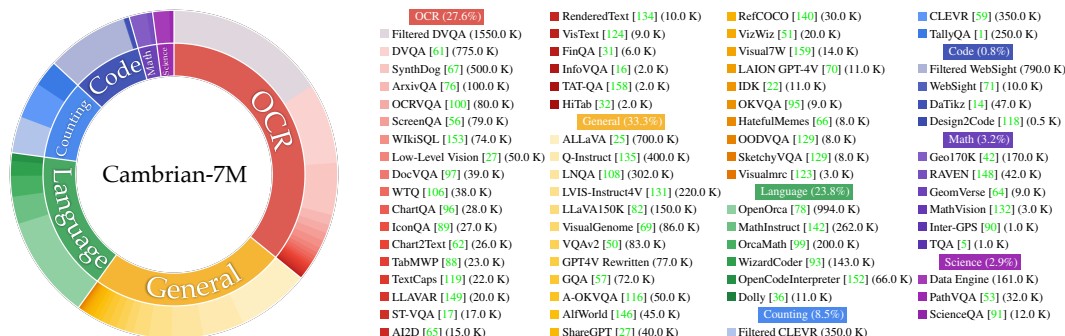

OCR (27.6%)
- Filtered DVQA (1550.0 K)
- DVQA [61] (775.0 K)
- SynthDog [67] (500.0 K)
- ArxivQA [76] (100.0 K)
- OCRVQA [100] (80.0 K)
- ScreenQA [56] (79.0 K)
- WIKiSQL [153] (74.0 K)
- Low-Level Vision [27] (50.0 K)
- DocVQA [97] (39.0 K)
- WTQ [106] (38.0 K)
- ChartQA [96] (28.0 K)
- IconQA [89] (27.0 K)
- Chart2Text [62] (26.0 K)
- TabMWP [88] (23.0 K)
- TextCaps [119] (22.0 K)
- LLAVAR [149] (20.0 K)
- ST-VQA [17] (17.0 K)
- AI2D [65] (15.0 K)
- RenderedText [134] (10.0 K)
- VisText [124] (9.0 K)
- FinQA [31] (6.0 K)
- InfoVQA [16] (2.0 K)
- TAT-QA [158] (2.0 K)
- HiTab [32] (2.0 K)

General (33.3%)
- ALLaVA [25] (700.0 K)
- Q-Instruct [135] (400.0 K)
- LNQA [108] (302.0 K)
- LVIS-Instruct4V [131] (220.0 K)
- LLaVA150K [82] (150.0 K)
- VisualGenome [69] (86.0 K)
- VQAv2 [50] (83.0 K)
- GPT4V Rewritten (77.0 K)
- GQA [57] (72.0 K)
- A-OKVQA [116] (50.0 K)
- AlfWorld [146] (45.0 K)
- ShareGPT [27] (40.0 K)
- RefCOCO [140] (30.0 K)
- VizWiz [51] (20.0 K)
- Visual7W [159] (14.0 K)
- LAION GPT-4V [70] (11.0 K)
- IDK [22] (11.0 K)
- OKVQA [95] (9.0 K)
- HatefulMemes [66] (8.0 K)
- OODVQA [129] (8.0 K)
- SketchyVQA [129] (8.0 K)
- Visualmrc [123] (3.0 K)

Language (23.8%)
- OpenOrca [78] (994.0 K)
- MathInstruct [142] (262.0 K)
- OrcaMath [99] (200.0 K)
- WizardCoder [93] (143.0 K)
- OpenCodeInterpreter [152] (66.0 K)
- Dolly [36] (11.0 K)

Counting (8.5%)
- Filtered CLEVR (350.0 K)

CLEVR [59] (350.0 K)
- TallyQA [1] (250.0 K)

Code (0.8%)
- Filtered WebSight (790.0 K)
- WebSight [71] (10.0 K)
- DaTikz [14] (47.0 K)
- Design2Code [118] (0.5 K)

Math (3.2%)
- Geo170K [42] (170.0 K)
- RAVEN [148] (42.0 K)
- GeomVerse [64] (9.0 K)
- MathVision [132] (3.0 K)
- Inter-GPS [90] (1.0 K)
- TQA [5] (1.0 K)

Science (2.9%)
- Data Engine (161.0 K)
- PathVQA [53] (32.0 K)
- ScienceQA [91] (12.0 K)

Figure 8: **Cambrian-7M: A Large-Scale Curated Instruction Tuning Dataset for MLLM. Left:** The inner circle shows the original distribution of Cambrian-10M. The outer circle shows the curated Cambrian-7M. **Right:** All the data sources in the Cambrian dataset as well as the ones filtered in data curation.

|  | Average | General | Knowledge | OCR&Chart | Vision-Centric |
|---|---|---|---|---|---|
| 150k | 53.7 | 68.0 | 51.3 | 45.2 | 50.5 |
| 250k | **54.3** | **68.1** | 51.5 | 45.3 | 52.2 |
| 350k | 54.3 | 67.4 | 51.4 | **46.0** | **52.3** |
| 450k | 54.2 | 68.0 | **52.2** | 45.5 | 50.7 |

Table 2: $t$ value between 250k and 350k obtains better performance.

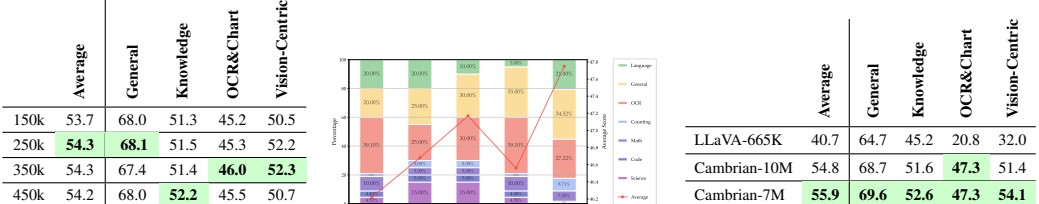

Figure 9: Exploring instruction tuning data mixture ratios.

|  | Average | General | Knowledge | OCR&Chart | Vision-Centric |
|---|---|---|---|---|---|
| LLaVA-665K | 40.7 | 64.7 | 45.2 | 20.8 | 32.0 |
| Cambrian-10M | 54.8 | 68.7 | 51.6 | **47.3** | 51.4 |
| Cambrian-7M | **55.9** | **69.6** | **52.6** | **47.3** | **54.1** |

Table 3: Performance improves with better instruction tuning data curation.

a limited number of learnable queries via global cross-attention. We also compare SVA with other connectors in Appendix H and show clear advantages.

> *Finding 8:* Spatial inductive bias and deep interaction between LLM and vision feature help to better aggregate and condense vision features.

## 4 Instruction Tuning Data for Training MLLMs

### 4.1 Data Collection

**Collecting Instruction Tuning Data from existing data sources** Unlike language data, multimodal (visual) instruction-tuning data is much rarer and harder to collect. To address this, we use existing multimodal benchmarks and datasets involving visual interaction data, such as Visual Question Answering (VQA) and OCR data. To help maintain conversational abilities [147], we also collect a small volume of high-quality language-only instruction-following data. We categorize data into General conversation, OCR, Counting, Code, Math, Science, and Language-only data. We list the data sources in Fig. 8, and the details of data preparation in Appendix G.

**Targeted Internet Data Collection Engine** As observed in Fig. 8, there is an unbalanced distribution of data. Some categories, such as science, have very few data sources, and each source has limited samples. Inspired by previous works [73], we introduce a data engine to create large-scale, reliable, high-quality knowledge-based instruction tuning data (see Fig. 15). Details are in Appendix G.3. Our data engine produces a large volume of reliable scientific data, increasing the diversity in the data pool. We generate 161k science-related data points—400% more than the previous combined data sources.

**Cambrian-10M** We create a large pool of instruction tuning data, which we refer to as Cambrian-10M. This pool contains approximately 9784k data points, offering a diverse range of data for our work and future research. We visualize its composition in Fig. 8.

### 4.2 Data Curation

Cambrian-10M is a large pool of instruction tuning data sourced from a variety of data sources, with an unbalanced data ratio between categories. Here, we take a preliminary step to study data curation by improving data balancing and adjusting data ratios.

| Model / Method | # Vis Tok. | General | | | | | Knowledge | | | | | OCR & Chart | | | | | Vision-Centric | | | | |
|---|---|---|---|---|---|---|---|---|---|---|---|---|---|---|---|---|---|---|---|---|---|
| | | Avg | $MME^P$ | MMB | $SEED^I$ | GQA | Avg | $SQA^I$ | $MMMU^V$ | $MathVista^M$ | AI2D | Avg | ChartQA | OCRBench | TextVQA | DocVQA | Avg | MMVP | RealworldQA | $CV\text{-}Bench^{2D}$ | $CV\text{-}Bench^{3D}$ |
| GPT-4V | UNK. | 63.0 | 1409.4 | 75.8 | 69.1 | 36.8 | 65.2 | 75.7 | 56.8 | 49.9 | 78.2 | 77.4 | 78.5 | 64.5 | 78.0 | 88.4 | 62.4 | 50.0 | 61.4 | 64.3 | 73.8 |
| Gemini-1.0 Pro | UNK. | - | 1496.6 | 73.6 | 70.7 | - | - | 79.5 | 47.9 | 45.2 | - | - | - | 65.9 | - | - | - | - | - | - | - |
| Gemini-1.5 Pro | UNK. | - | - | - | - | - | - | - | 58.5 | 52.1 | 80.3 | - | 81.3 | - | 73.5 | 86.5 | - | - | 67.5 | - | - |
| Grok-1.5 | UNK. | - | - | - | - | - | - | - | 53.6 | 52.8 | 88.3 | - | 76.1 | - | 78.1 | 85.6 | - | - | 68.7 | - | - |
| MM-1-8B | 144 | - | 1529.3 | 72.3 | 69.9 | - | - | 72.6 | 37.0 | 35.9 | - | - | - | - | - | - | - | - | - | - | - |
| MM-1-30B | 144 | - | 1637.6 | 75.1 | 72.1 | - | - | 81.0 | 44.7 | 39.4 | - | - | - | - | - | - | - | - | - | - | - |
| *Base LLM: Llama-3-Ins-8B* | | | | | | | | | | | | | | | | | | | | | |
| Mini-Gemini-HD-8B | 2880 | 72.7 | **1606.0** | 72.7 | 73.2 | 64.5 | 55.7 | 75.1 | 37.3 | 37.0 | 73.5 | 62.9 | 59.1 | 47.7 | 70.2 | 74.6 | 51.5 | 18.7 | 62.1 | 62.2 | 63.0 |
| LLaVA-NeXT-8B | 2880 | 72.5 | 1603.7 | 72.1 | 72.7 | **65.2** | 55.6 | 72.8 | 41.7 | 36.3 | 71.6 | 63.9 | 69.5 | 49.0 | 64.6 | 72.6 | 56.6 | 38.7 | 60.1 | 62.2 | 65.3 |
| Cambrian-1-8B | 576 | **73.1** | 1,547.1 | **75.9** | **74.7** | 64.6 | **61.3** | **80.4** | 42.7 | **49.0** | 73.0 | **71.3** | **73.3** | **62.4** | **71.7** | **77.8** | **65.0** | **51.3** | **64.2** | **72.3** | **72.0** |
| *Base LLM: Vicuna-1.5-13B* | | | | | | | | | | | | | | | | | | | | | |
| Mini-Gemini-HD-13B | 2880 | 70.7 | 1597.0 | 68.6 | 70.6 | 63.7 | 54.1 | 71.9 | 37.3 | 37.0 | 70.1 | 60.8 | 56.6 | 46.6 | 70.2 | 69.8 | 49.4 | 19.3 | 57.5 | 53.6 | 67.3 |
| LLaVA-NeXT-13B | 2880 | 69.9 | 1575.0 | 70.0 | 65.6 | **65.4** | 53.7 | 73.5 | 36.2 | 35.1 | 70.0 | 62.9 | 62.2 | 51.4 | 67.1 | 70.9 | 55.9 | 36.0 | 59.1 | 62.7 | 65.7 |
| Cambrian-1-13B | 576 | **73.7** | **1,610.4** | **75.7** | **74.4** | 64.3 | **60.2** | **79.3** | **40.0** | **48.0** | **73.6** | **71.3** | **73.8** | **61.9** | **72.8** | **76.8** | **62.2** | **41.3** | **63.0** | **72.5** | **71.8** |
| *Base LLM: Hermes2-Yi-34B* | | | | | | | | | | | | | | | | | | | | | |
| Mini-Gemini-HD-34B | 2880 | 76.2 | 1659.0 | 80.6 | 75.3 | 65.8 | 62.4 | 77.7 | 48.0 | 43.4 | **80.5** | 68.1 | 67.6 | 51.8 | 74.1 | **78.9** | 63.8 | 37.3 | 67.2 | 71.5 | 79.2 |
| LLaVA-NeXT-34B | 2880 | 76.0 | 1633.2 | 79.3 | **75.9** | **67.1** | 62.5 | 81.8 | 46.7 | 46.5 | 74.9 | 67.7 | 68.7 | 54.5 | 69.5 | 78.1 | 64.0 | 47.3 | 61.0 | 73.0 | 74.8 |
| Cambrian-1-34B | 576 | **76.8** | **1689.3** | **81.4** | 75.3 | 65.8 | **67.0** | **85.6** | **49.7** | **53.2** | 79.7 | **71.9** | **75.6** | **60.0** | 76.7 | 75.5 | **68.5** | **52.7** | **67.8** | **74.0** | **79.7** |

Table 4: **Comparison of Cambrian-1 with other leading MLLMs.** Cambrian-1 outperforms other open-source models and achieves competitive performance, compared to proprietary models such as GPT-4V, Gemini, and Grok-1.5. Despite using only 576 visual tokens, Cambrian-1 performs better on OCR & Chart and Vision-Centric benchmarks compared to Mini-Gemini-HD and LLaVA-NeXT, which use 2880 tokens.

**Data Balancing** We follow previous work [109, 138] to set thresholds $t$ for the number of data points from a single data source. We choose $t = 150k$, $250k$, $350k$, and $450k$ in this section and observe an elbow effect in Table 2—finding that a threshold between $250k$ and $350k$ work the best for Cambrian-10M. We also plot in Appendix G.4 the cumulative sum of counts for entries sorted by counts from tail to head and we see this intermediate threshold prevents explosive heavy tail.

**Data Ratio** Cambrian-10M is designed for visual instruction tuning. Given the various capabilities of different types of data, it is essential to balance the ratio of these data types. We conduct pilot experiments with a fixed dataset size of 1350k, examining the impact of different data ratios. We visualize the results in Fig. 9 and summarize our findings as follows: (i) Balancing General, OCR and Language data is crucial. (ii) Performance on knowledge-intensive tasks is influenced by multiple factors, often requiring a mix of OCR, chart, reasoning, and general perception.

**Cambrian-7M** By applying data filtering to Cambrian-10M with our identified data ratio, we create a smaller but higher-quality dataset called Cambrian-7M. Table 3 showcases the benefits of a well-balanced and carefully curated dataset. Despite having fewer samples, Cambrian-7M demonstrates improved performance. We additionally apply system prompts in Cambrian-7M to avoid the "answer machine phenomenon", see more details in Appendix G.2.

## 5 State of the Art Performance

Finally, we leverage the insights from all our previous studies to train a family of MLLMs we call Cambrian-1. We train models using LLM backbones of various scales: LLaMA-3-Instruct-8B [4], Vicuna-1.5-13B [151], and Hermes-2-Yi-34B [139]. Our vision component combines four models— OpenAI CLIP ViT-L/14@336, SigLIP ViT-SO400M/14@384, OpenCLIP ConvNeXt-XXL@1024, and DINOv2 ViT-L/14@518 (Section 2.5)—via the Spatial Vision Aggregator (Section 3). We pre-train the connector using 2.5M adapter data and instruction tune using our Cambrian-7M data mix (Section 4.2). Our models are evaluated on the benchmarks categorized in Section 2.1, with results presented in Table 4.[††]

Cambrian-1 surpasses open-source models like LLaVA-NeXT and Mini-Gemini. Thanks to the SVA, Cambrian-1 excels in tasks requiring high-resolution image processing, even with only 576 image tokens—about 1/5 of the tokens used by LLaVA-NeXT and Mini-Gemini. Cambrian-1 also achieves comparable performance to the best proprietary models, such as GPT-4V, Gemini-Pro, and MM-1, on several benchmarks. We provide model weights, open-source code, datasets, and detailed recipes for model training and evaluation. We hope our work will strengthen the open research community and accelerate research in both visual representation learning and multimodal systems.

---

[††]For the General Average, we note that GPT-4's performance on the GQA test set is low, possibly because other models are trained on the GQA training set, whereas the training set used for GPT-4 is unclear.

## Acknowledgements

We are grateful to LLaVA [82] for their excellent codebase, which served as the launching point for our research. Special thanks to Hexu Zhao for extensive discussions and knowledge-sharing around FSDP and large-scale training techniques, and to Jiasen Lu for helpful discussions on TPU and JAX distributed training infrastructure. We also appreciate the assistance and responses from the PyTorchXLA team via GitHub.

We are thankful to Kaiming He for early discussions on multi-modal large language models. We also thank Zhuang Liu, Junlin Han, Yuexiang Zhai, Tianzhe Chu, Daohan Lu, Weiyang Jin, Boyang Zhang, and Jiayi Pan for reviewing this manuscript. We also acknowledge DeepSeek [87] for the paper template inspiration.

This work was primarily supported by the Google TPU Research Cloud (TRC) program and the Google Cloud Research Credits program (GCP19980904). Additional support was provided by the NYU IT High Performance Computing resources, services, and staff expertise. S.X. would like to thank the OpenAI Researcher Access program, Open Path AI Foundation, and an Amazon Research award for their support. S.T. is supported by the OpenAI SuperAlignment Fellowship, and E.B. is supported by the NDSEG Fellowship.

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

# A  Broader Discussion

We advocate for using MLLMs as an interface to evaluate visual representations, as previous benchmarks are becoming saturated and do not adequately reflect the diverse and complex perception challenges of the real world. Our work highlights the current gap between language-supervised models and self-supervised learning models and demonstrates the potential of bridging this gap. However, it is known that features of language-supervised models behave like a *bag-of-words* [125, 144], underscoring the need for advancements in vision-only models to ensure better visual understanding. We hope to inspire future research into developing better vision-only models intended to be adapted into the MLLM setting, that more effectively leverage large-scale datasets [85] and preserve the advantages in visual grounding [126].

As we observe in Table 4, a well-trained open-source model such as Cambrian-1 can match or even outperform proprietary models on many existing benchmarks. However, the use and evaluation of MLLMs extend far beyond the current scope of benchmarks—to conversational ability, creativity, reliability, and overall user experience. Developing models solely based on benchmark results can result in an "answer machine", over-optimized for benchmarks but lacking in practical interaction capabilities. Therefore, the development of MLLMs that better align with human and societal needs is a continuously evolving process, both in terms of evaluation and model development.

Our current Cambrian-1 model uses a moderate number of visual tokens and does not adopt the any-resolution strategy [30, 77, 81] to handle ultra high-resolution images or those with extreme aspect ratios, which require a larger number of visual tokens. For specialized tasks like V*Bench [136], which require processing ultra high-resolution images, increasing the resolution and number of visual tokens could lead to an HD version of the Cambrian-1 model.

One promising direction for post-training alignment is through reinforcement learning rather than supervised fine-tuning. Many MLLM studies, including Cambrian, primarily focus on supervised fine-tuning. Yet, recent advancements in LLMs [38, 104, 110, 156] and some in MLLMs [141, 146] suggest that reinforcement learning from human or environmental feedback can further improve models, potentially surpassing the limits of supervised fine-tuning, especially in decision-making abilities.

Cambrian-10M (Fig. 8 and Section 4) provides a rich pool of data for studying data curation in fine-tuning MLLMs. Our work takes an initial step in curating higher-quality data to enable more efficient and effective instruction tuning. We believe there is room for further improvement in the data curation pipeline, and we hope this work can serve as a foundation for future research.

Additionally, training large-scale models requires careful design of model sharding, data sharding, and infrastructure adaptations. In this work, we train our model on TPU-V4 [60] with FSDP [150] using TorchXLA. We share our experiences, technical challenges, and solutions in Appendix C. We also open-source our implementation and provide tutorials to help the community undertake large-scale training more efficiently.

To conclude, Cambrian-1 introduces a family of state-of-the-art MLLM models that achieve top performance across diverse benchmarks and excel in visual-centric tasks. We provide model weights, open-source code, datasets, and detailed recipes for model training and evaluation. We hope our work will strengthen the open research community and accelerate future advancements in both visual representation learning and multimodal systems.

# B  Multimodal LLMs: Preliminaries and Related Work

The key components of MLLM research include the *Large Language Model*, *Visual Encoder*, *Multimodal Connector*, *Data Curation Pipeline*, *Instruction Tuning Strategy*, and *Evaluation & Benchmarking*. Each component has its intricacies, and understanding their interactions presents significant challenges. Our study investigates these aspects from a vision-centric perspective.

**Large Language Model**  Advanced LLMs [4, 101, 127, 128] are the foundation of an MLLM. After instruction-tuning on multimodal data, these models can be prompted to solve a variety of complex tasks and generate free-form responses leveraging input from a visual encoder. Recent MLLM research focuses on enhancing the LLM backbone [10, 75, 81], resulting in improved performance on benchmarks like MMMU [143] and AI2D [54]. However, this improvement raises the concern

that our current *multimodal* evaluation is biased by the development of LLMs, neglecting a true assessment of visual perception. For example, some benchmarks such as MMMU [143] are dominated by LLM capabilities, underscoring the need for evaluations that genuinely assess multimodality (see Section 2.1).

**Visual Encoder**  Most MLLMs utilize language-supervised models like CLIP [109, 122, 145], which benefit from the massive scale of noisy web image-text data. However, there is a much broader pool of visual models that learn representations using only visual signals—such as self-supervised models [9, 103], segmentation [68], depth-supervised [15], and diffusion models [74, 112] (see Fig. 2). Recent work [87, 126] advocates for incorporating these diverse vision models into MLLMs. In this study, we systematically examine the impact of various vision backbones on MLLM performance (Section 1) and explore the benefits of model ensembles (Section 2.5).

**Multimodal Connector**  Representations from a visual encoder cannot be natively processed by an LLM—they must be mapped into the LLM token space by a *connector*. There are three primary approaches to connector design: Resamplers [6], Q-Formers [11, 37], and MLP Projectors [43, 80, 82, 157]. We begin our exploration using an MLP projector, which is highly effective but presents challenges: the visual token count grows quadratically with image resolution, inhibiting scaling context length input resolution. For example, LLaVA-Next [81] requires 2880 visual tokens to process one 672px image. To address this, we explore new vision connector designs that process high-resolution images while maintaining a smaller number of visual tokens (Section 3).

**Instruction Tuning Data**  Visual instruction tuning data is crucial but hard to collect, as it rarely naturally exists on the internet. Previous work [37, 80, 98] transforms existing VQA benchmarks [50, 69] into instruction tuning data, showing marked MLLM performance improvements. With this inspiration, we collect all VQA benchmarks and visual interaction data that we can find (Fig. 8), study data balancing and category mixtures (Section 4.2), and develop an internet data collection engine to fill in the gaps (Section 4.1).

**Instruction Tuning**  Most current MLLMs leverage pre-trained LLMs and visual encoders, fine-tuning the LLM and connector using visual instruction tuning data. Some aspects of the tuning recipe are up for debate, including whether to pre-train the connector before joint fine-tuning with the LLM , and whether to freeze or unfreeze the vision encoder during fine-tuning [63, 98]. Additionally, some recent proprietary models explore end-to-end training from scratch [49, 102]. In this work, we use pre-trained models and revisit the debated recipe aspects with extensive studies, providing more insights for future MLLM research (Section 2.3).

**Evaluation & Benchmarking**  There is an extensive set of benchmarks that evaluate various aspects of MLLMs, such as perception [45, 83], knowledge [91, 92], chart interpretation [84, 96], and visual capabilities [126, 136]. Instead of over-optimizing for specific benchmarks, we advocate for examining aggregates of benchmarks that focus on specific capabilities. To achieve this, we analyze existing benchmarks, categorize them, and assess the extent to which they measure *multimodality* (Section 2.1). Additionally, we find there are currently few benchmarks focused on vision-centric evaluation, and those that do exist contain relatively few images, leading to higher variance during evaluation. To address this issue, we propose a new vision-centric benchmark by reformulating classic vision tasks (Section 2.2).

## C  Training, Infrastructure, and Implementation

All models in this paper were trained using TPU-V4 pods [60]; we evaluate using NVIDIA A6000, A100, and H100 cards. The experiments in Section 2.4 require less than 24 hours on a TPU-V4-128, while our final Cambrian-1 models are trained in less than 4 days on a TPU-V4-512.

To enable and facilitate large-scale parallel training on TPUs, we employ TorchXLA with FSDP [150] to handle training sharding and parallelism. Training a large-scale multimodal model with TorchXLA on TPU is a challenging journey, as there are no open-source codebases and many critical features are not supported in the TorchXLA or TorchXLA FSDP libraries. To provide a brief taste of the difficulties: TPUs require a static graph throughout the program, which requires ground-up rewrites of dynamically-written open-source PyTorch codebases; model resuming is not implemented in TorchXLA, which is especially crucial when training on preemptable TPUs; existing TorchXLA FSDP tutorials fail to compile due to version changes in TorchXLA, updates in Hugging Face

Transformers & Accelerate, or simply inherent issues with the tutorial; loading very large models (over 30 billion parameters) with the TorchXLA FSDP library is natively impossible due to the 100GB memory constraints of TPU-V4s, and requires extensive workarounds.

To this end, we have rewritten or developed many new functions to make this research possible. For instance, we rewrote the TorchXLA FSDP Sharding API to load very large models; we implemented model resuming on TorchXLA; we rewrote parts of the Hugging Face Transformers FSDP and gradient checkpointing implementations to enable large-scale FSDP training. We are committed to open-sourcing our codebase and publishing a comprehensive tutorial to share our insights, with the hope of inspiring and supporting future research and open-source contributions to the TPU and TorchXLA ecosystem.

# D  Analyzing the Benchmarks

**MLLM Benchmark Performance Confusion Matrix**

We evaluate the benchmark scores for our one-stage, two-stage finetune-only and hybrid models, and then plot the correlation matrix for the pool of MLLM benchmarks. The correlation plot displays in Fig. 10. The result demonstrates that MMMU is less correlated in measuring model performance to other benchmarks. Nonetheless, we acknowledge it is widely used and therefore cluster it into knowledge-based QAs based on the nature of their questions.

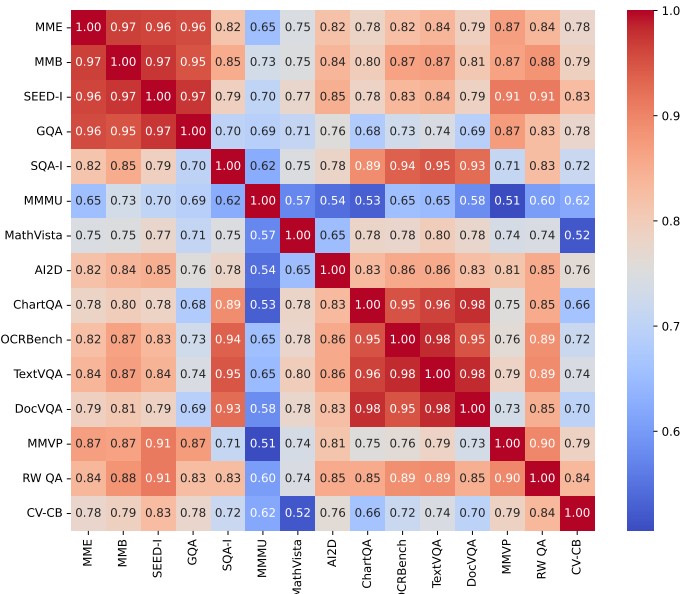

Figure 10: **Correlation matrix for MLLM benchmarks.** The correlation matrix for MLLM benchmarks with respect to different vision backbones. The correlation matrix helps us to analysis and group benchmarks.

# E  Cambrian Vision-Centric Benchmark (CV-Bench)

**CV-Bench Curation**  Below we describe the procedure for programmatically constructing questions for each task. To ensure reliability, we also *manually inspect each question*, removing those that are unclear, ambiguous, or erroneous.

*Spatial Relationship (2D).*  We consider images with two distinct ground-truth object categories and use visual prompts (bounding boxes) to avoid ambiguity when multiple instances are present. In these questions, we designate an anchor object, and the question asks for the direction of the other object relative to this anchor.

*Object Counting (2D).*  This tests the model's ability to count objects. When generating options for these questions, we construct multiple-choice options that are similar to the correct answer. For

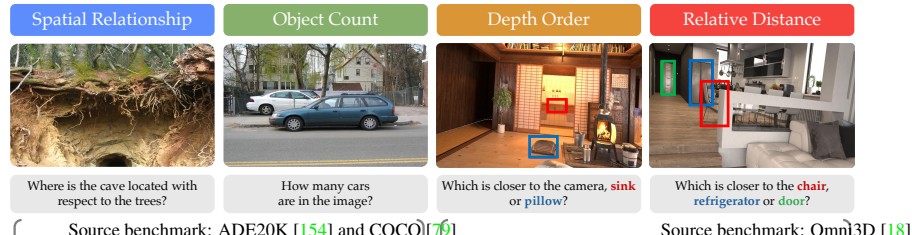

Figure 11: **Cambrian Vision-Centric Benchmark (CV-Bench).** We repurpose standard vision benchmarks to evaluate the fundamental 2D and 3D visual understanding of MLLMs. See Section 2.2 for more details.

| Type | Task | Description | Sources | # Samples |
|------|------|-------------|---------|-----------|
| 2D | **Spatial Relationship** | Determine the relative position of an object w.r.t. the anchor object. Consider left-right or top-bottom relationship. | ADE20K COCO | 650 |
| | **Object Count** | Determine the number of instances present in the image. | ADE20K COCO | 788 |
| 3D | **Depth Order** | Determine which of the two distinct objects is closer to the camera. | Omni3D | 600 |
| | **Relative Distance** | Determine which of the two distinct objects is closer to the anchor object. | Omni3D | 600 |

Table 5: Breakdown of the 2D and 3D tasks evaluated in the Cambrian Vision-Centric Benchmark (CV-Bench). The examples are sourced from ADE20K [154], COCO [79], and Omni3D [18].

example, if the correct answer is 4, the options might be 2, 3, 4, 5, & 6. We also include existence check examples where the correct count is 0.

*Depth Order (3D).* We consider images with two distinct categories (i.e., object A and object B) and use visual prompts (e.g., bounding boxes with two different colors) to avoid ambiguity. We define "closer" as follows: object A is closer to the camera than object B only if the farthest vertex of object A is closer[‡‡] to the camera than the nearest vertex of object B by a specified offset.

*Relative Distance (3D).* We consider images with three distinct categories (i.e., anchor, object A, and object B), and use visual prompts (e.g., bounding boxes with three different colors) to avoid ambiguity. Object A is closer than object B only if the farthest distance from A's vertices is shorter than the shortest distance from B's vertices to the anchor object by a certain offset.

**Curation Procedure** We provide an overview of the data curation process in Fig. 12, which is conducted in a semi-automatic manner. The procedure consists of two main steps:

First, using the original benchmarks and their associated ground truth annotations, we generate query and answer pairs. These pairs are tailored to specific tasks: 2D-related tasks with COCO and ADE20K datasets, and 3D-related tasks with Omni3D.

Second, after generating the query and answer pairs, we engage human experts to manually filter out any incorrect or ambiguous queries to enhance the quality of benchmark. Each query is assigned one of three statuses: accepted (used as is), modified (where the incorrect answer is modified), and rejected (queries that are ambiguous, such as those too small or difficult to discern, even for human experts).

Following this two-stage process, we finalize the benchmark, which results in a total of 2638 image queries with improved accuracy and reliability. Subsequently, we will discuss the methods of human verification and the evaluation metrics used in this process.

**Human verification**

---

[‡‡]We use the Euclidean distance.

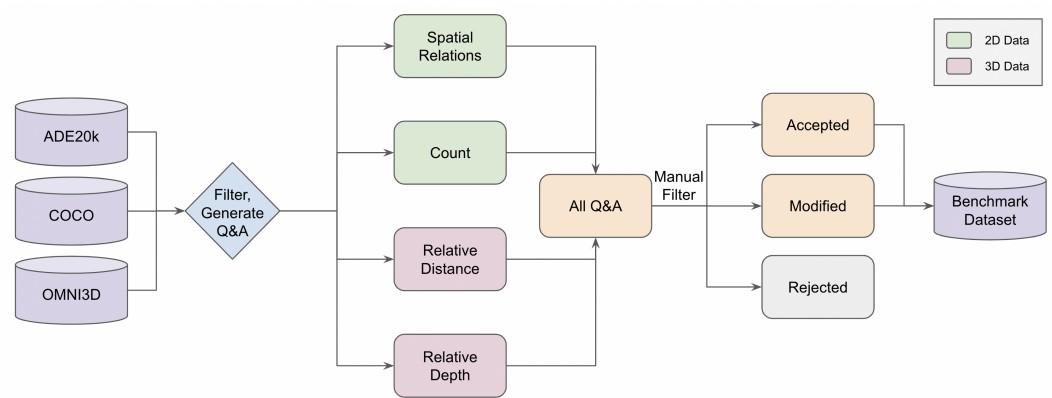

Figure 12: **CV-CB Benchmark Filtering.** We reformulate classic 2D and 3D CV benchmarks into Q&A questions to evaluate MLLM's visual capabilities.

There are multiple reasons for the above generated data to be inaccurate. One of the main reasons is sparse annotations, but occasionally there could be wrong annotations as well.

Thus, we need manual inspection to change/remove these examples generated. Here are a few criteria we followed while manually filtering COCO and ADE20k data.

For **Counting** question types, if all instances of a category are not annotated, the ground truth would have lower count than the actual number of instances appearing in the image. In a few cases where the image distinctly has different countable instances of the object, we change the options/answer. In case the count is ambiguous, we reject the data sample altogether.

For **Relative Distance** question types without annotation, if the question is asked about two objects A and B and if there are two instances of a specific category (say A), the relative location of A w.r.t B can be have multiple correct answers. We reject the sample in this case. We also reject cases with clear incorrect annotations.

**Benchmark Evaluation**  To ensure that equal importance is given to both 2D and 3D tasks, we use an evaluation metric that is the average of the accuracies obtained from these tasks. Specifically, the overall performance is calculated as follows:

$$\text{Accuracy}_{2D} = \left( \frac{\text{Accuracy}_{COCO} + \text{Accuracy}_{ADE20k}}{2} \right)$$

$$\text{Overall Accuracy} = \left( \frac{\text{Accuracy}_{2D} + \text{Accuracy}_{3D}}{2} \right)$$

## F  Vision Models in MLLMs

As mentioned in Section 2.4, we use MLLM as an interface to evaluate vision model's different capabilities. Here, we list details in terms of the model selection, full results, and data split.

### F.1  Details of Vision Models

In our exploration of versatile vision models, we select thirteen models and group them into four categories: *language-supervised models* (i.e., OpenAI CLIP [109], SigLIP [145], DFN-CLIP [40], EVA-CLIP [122] and OpenCLIP [33]), *self-supervised models* (i.e., DINOv2 [103], I-JEPA [9], MAE [52], MoCo v3 [29]), *class-supervised models* (ImagetNet22K ViT [39]) and *other models* such as stable diffusion [112], segmentation models like SAM [68], and depth estimation models like MiDaS [15]. To provide a clear understanding of the specific variant evaluated, we meticulously detail their backbone architectures, resolution, number of tokens, and hidden dimension sizes in Table 6. For models that output a large number of patches in the last layer (e.g., SAM and ConvNeXt) we interpolate to the number of tokens specified in Table 6, and denote interpolation with [I].

---

We extract features following the practice in [12]

| Supervision Type | Method | Architecture | Patch Size | Res. | # Tok. | Hidden Size |
|---|---|---|---|---|---|---|
| Language-Supervised | | | | | | |
| Language | OpenAI CLIP | ViT-L | 14 | 336 | 576 | 768 |
| | DFN-CLIP | ViT-L | 14 | 224 | 256 | 1024 |
| | DFN-CLIP | ViT-H | 14 | 378 | 729 | 1280 |
| | EVA-CLIP-02 | ViT-L | 14 | 336 | 576 | 1024 |
| | SigLIP | ViT-L | 16 | 384 | 576 | 1024 |
| | SigLIP | ViT-SO400M | 14 | 384 | 729 | 1152 |
| | OpenCLIP | ConvNeXT-L | - | 512 | [1]576 | 1536 |
| | OpenCLIP | ConvNeXT-L | - | 1024 | [1]576 | 1536 |
| | OpenCLIP | ConvNeXT-XXL | - | 1024 | [1]576 | 3072 |
| Self-Supervised | | | | | | |
| Contrastive | DINOv2 | ViT-L | 14 | 336 | 576 | 1024 |
| | DINOv2 | ViT-L | 14 | 518 | [1]576 | 1024 |
| | MoCo v3 | ViT-B | 16 | 224 | 196 | 768 |
| | MoCo v3 | ViT-L | 16 | 224 | 196 | 1024 |
| Masked | MAE | ViT-L | 16 | 224 | 196 | 1024 |
| | MAE | ViT-H | 14 | 224 | 256 | 1280 |
| JEPA | I-JEPA | ViT-H | 14 | 224 | 256 | 1280 |
| Other | | | | | | |
| Segmentation | SAM | ViT-L | 16 | 1024 | [1]576 | 1024 |
| | SAM | ViT-L | 16 | 1024 | [1]576 | 1280 |
| Depth | MiDaS 3.0 | ViT-L | 16 | 384 | 576 | 1024 |
| | MiDaS 3.1 | ViT-L | 16 | 518 | 1024 | 1024 |
| Diffusion | Stable Diffusion 2.1 | VAE+UNet | 16 | 512 | 1024 | 3520 |
| Class Labels | SupViT | ViT-L | 16 | 224 | 196 | 1024 |
| | SupViT | ViT-H | 14 | 224 | 256 | 1280 |

Table 6: Catalog of all vision backbones tested. [1] denotes that the visual tokens have been interpolated down to the specified length.

| Method | Architecture | Patch Size | Resolution | # Tokens | Linear Probing (%) |
|---|---|---|---|---|---|
| EVA-CLIP-02 | ViT-L | 14 | 336 | 576 | 85.0 |
| DFN-CLIP | ViT-L | 14 | 224 | 256 | 83.6 |
| DINOv2 | ViT-L | 14 | 336 | 576 | 83.1 |
| OpenCLIP | ConvNeXt-L | - | 512 | 576 | 82.9 |
| OpenAI CLIP | ViT-L | 14 | 336 | 576 | 80.3 |
| I-JEPA | ViT-H | 14 | 224 | 256 | 77.0 |
| Supervised | ViT-L | 16 | 224 | 196 | 74.5 |
| MoCo v3 | ViT-B | 16 | 224 | 196 | 71.9 |
| MiDaS | ViT-L | 16 | 384 | 576 | 70.1 |
| MAE | ViT-L | 16 | 224 | 196 | 68.3 |

Table 7: Linear Probing Results of Different Vision Backbones

## F.2 Full Results of Different Vision Backbones

Here, we also show a ranking version of Fig. 5. We observe a clear advantage of CLIP models over non-CLIP models. We also observe that within the family of CLIP models, each model perform differently in different domains. This provide insight into both vision model development and data curation in training large vision models.

For the above-listed vision models in Table 6, they are integrated as the vision encoder of the MLLMs. These MLLMs are trained on various adapter adapter data splits (i.e., 0, 0.5 and 1.2 million), and subsequently fine-tuned on a 737K instruction tuning dataset provided in LLaVA-1.5[80]. For the adapter data splits, the 0M split indicates that no initial adapter pertaining phase is employed for the

| **Language Supervised** | | | | | | | **Self-Supervised & Other** | | | | | | |
|---|---|---|---|---|---|---|---|---|---|---|---|---|---|
| Model | Architecture | All | G | K | O | V | Model | Architecture | All | G | K | O | V |
| SigLIP | ViT-SO400M/14@384 | 1 | 1 | 1 | 2 | 1 | DINOv2 | ViT-L/14@518 | 1 | 1 | 1 | 1 | 1 |
| OpenCLIP | ConvNeXt-XXL@1024 | 2 | 6 | 8 | 1 | 3 | DINOv2 | ViT-L/14@336 | 2 | 2 | 3 | 3 | 2 |
| DFN-CLIP | ViT-H/14@378 | 3 | 4 | 2 | 5 | 4 | MAE | ViT-L/14@224 | 3 | 5 | 2 | 2 | 4 |
| OpenCLIP | ConvNeXt-L@1024 | 4 | 8 | 7 | 3 | 8 | I-JEPA | ViT-H/14@224 | 4 | 3 | 6 | 8 | 3 |
| SigLIP | ViT-L/16@384 | 5 | 5 | 4 | 4 | 6 | SD2.1 | VAE+UNet/16@512 | 5 | 7 | 9 | 9 | 5 |
| OpenAI CLIP | ViT-L/14@336 | 6 | 3 | 6 | 6 | 7 | MiDaS 3.0 | ViT-L/16@384 | 6 | 6 | 8 | 5 | 6 |
| EVA-CLIP-02 | ViT-L/14@336 | 7 | 2 | 5 | 8 | 2 | SupViT | ViT-L/16@224 | 7 | 4 | 9 | 4 | 8 |
| OpenCLIP | ConvNeXt-L@512 | 8 | 7 | 3 | 7 | 9 | MoCo v3 | ViT-B/16@224 | 8 | 8 | 4 | 7 | 7 |
| DFN-CLIP | ViT-L/14@224 | 9 | 9 | 9 | 9 | 10 | MoCo v3 | ViT-L/16@224 | 9 | 9 | 5 | 6 | 9 |
| DINOv2* | ViT-L/14@518 | 10 | 10 | 10 | 10 | 5 | SAM | ViT-H/16@1024 | 10 | 10 | 10 | 10 | 10 |

Table 8: Benchmark performance rankings for MLLMs built upon language-supervised and self-supervised vision encoders across all benchmarks (All), and across general (G), knowledge (K), OCR & chart (O), and vision-centric (V) benchmark categories. Full results for all models on each benchmark are tabulated in Table 11. *We add DINOv2 here to show its standing amongst the CLIP models.

MLLM. The 0.5M data split utilizes the 558K adapter data from LLaVA-1.5[80], while the 1.2M variant uses ShareGPT4V-PT dataset [27].

**0M Adapter Data + 737K Instruction Tuning Data**   As shown in Table 9, we provide 20 results for different variants of the above-mentioned thirteen vision backbones. Among them, language-supervised models show superior performance. Especially, OpenCLIP ConvNeXT-XXL@1024 model surpasses all other models on DocVQA with over 12%, indicating its potential to handle OCR-related benchmarks.

| | Vision Backbone | | General | | | | Knowledge | | | | OCR & Chart | | | | Vision-Centric | | | |
|---|---|---|---|---|---|---|---|---|---|---|---|---|---|---|---|---|---|---|
| Model | Architecture | Average | $MME^P$ | MMB | $SEED^I$ | GQA | $SQA^I$ | $MMMU^V$ | $MathVista^M$ | AI2D | ChartQA | OCRBench | TextVQA | DocVQA | MMVP | RealWorldQA | $CV\text{-}Bench^{2D}$ | $CV\text{-}Bench^{3D}$ |
| **Language Supervised** | | | | | | | | | | | | | | | | | | |
| OpenAI CLIP | ViT-L/14@336 | 48.37 | 1,419.43 | 61.45 | 59.85 | 62.26 | 69.87 | 34.50 | 27.80 | 59.82 | 33.73 | 31.70 | 55.39 | 28.00 | 11.33 | 53.46 | 56.44 | 57.40 |
| DFN-CLIP | ViT-L/14@224 | 38.78 | 1,172.50 | 49.53 | 49.74 | 52.94 | 67.74 | 34.00 | 27.30 | 56.99 | 16.75 | 4.87 | 44.81 | 11.19 | 6.67 | 46.97 | 40.29 | 52.08 |
| DFN-CLIP | ViT-H/14@378 | 36.79 | 1,091.76 | 41.28 | 44.32 | 50.48 | 65.54 | 33.65 | 26.50 | 56.76 | 15.56 | 2.70 | 43.41 | 10.15 | 4.67 | 47.06 | 39.07 | 52.91 |
| EVA-CLIP-02 | ViT-L/14@336 | 45.84 | 1,325.17 | 58.21 | 62.99 | 62.03 | 68.67 | 35.00 | 27.50 | 58.26 | 19.40 | 22.50 | 51.08 | 16.36 | 24.67 | 52.68 | 52.98 | 54.83 |
| SigLIP | ViT-L/16@384 | 48.80 | 1,383.42 | 61.02 | 63.56 | 61.85 | 68.91 | 35.29 | 29.70 | 57.87 | 34.96 | 29.60 | 56.73 | 28.31 | 23.33 | 52.68 | 52.95 | 54.83 |
| SigLIP | ViT-SO400M/14@384 | 47.57 | 1,376.75 | 58.76 | 60.59 | 60.92 | 69.01 | 34.40 | 26.50 | 58.35 | 30.72 | 28.60 | 55.10 | 28.31 | 19.33 | 50.71 | 52.33 | 58.67 |
| OpenCLIP | ConvNeXt-L@512 | 47.38 | 1,404.01 | 57.62 | 61.90 | 60.34 | 69.06 | 33.90 | 29.10 | 58.39 | 28.04 | 25.20 | 55.45 | 28.41 | 24.00 | 54.12 | 53.46 | 48.91 |
| OpenCLIP | ConvNeXt-L@1024 | 39.02 | 1,139.60 | 14.64 | 49.59 | 37.91 | 65.71 | 34.30 | 27.30 | 54.13 | 32.97 | 12.05 | 52.61 | 38.36 | 9.67 | 47.45 | 52.68 | 38.04 |
| OpenCLIP | ConvNeXt-XXL@1024 | 41.83 | 1,219.47 | 48.00 | 49.88 | 55.09 | 66.14 | 35.69 | 27.60 | 56.67 | 16.92 | 5.00 | 46.90 | 40.98 | 16.00 | 47.32 | 43.40 | 52.75 |
| **Self Supervised** | | | | | | | | | | | | | | | | | | |
| DINOv2 | ViT-L/14@336 | 41.18 | 1,262.66 | 49.62 | 56.80 | 60.30 | 65.10 | 35.00 | 26.40 | 56.41 | 16.48 | 3.10 | 44.04 | 11.90 | 18.67 | 50.20 | 49.43 | 52.25 |
| DINOv2 | ViT-L/14@518 | 40.60 | 1,242.48 | 51.00 | 53.39 | 60.38 | 64.55 | 34.50 | 26.20 | 57.53 | 15.11 | 2.90 | 44.28 | 10.95 | 14.00 | 48.63 | 46.13 | 57.90 |
| MoCo v3 | ViT-B/16@224 | 34.94 | 966.45 | 36.77 | 33.00 | 47.35 | 62.96 | 32.80 | 26.50 | 55.05 | 16.04 | 2.60 | 43.81 | 10.31 | 6.67 | 45.36 | 39.03 | 52.83 |
| MoCo v3 | ViT-L/16@224 | 34.70 | 1010.18 | 34.64 | 41.71 | 47.46 | 64.70 | 33.70 | 26.30 | 55.05 | 16.24 | 2.70 | 42.60 | 10.39 | 4.00 | 45.36 | 44.67 | 35.16 |
| MAE | ViT-L/14@224 | 37.69 | 1,114.07 | 42.30 | 35.93 | 55.20 | 63.51 | 34.60 | 26.00 | 56.10 | 16.11 | 2.70 | 43.63 | 10.83 | 14.00 | 44.80 | 45.81 | 55.75 |
| MAE | ViT-H/14@224 | 38.58 | 1,083.35 | 41.15 | 50.99 | 60.34 | 63.49 | 34.30 | 26.00 | 56.49 | 15.63 | 3.20 | 43.98 | 11.00 | 12.00 | 46.30 | 47.18 | 54.90 |
| I-JEPA | ViT-H/14@224 | 38.88 | 1,132.07 | 44.68 | 51.74 | 55.37 | 66.04 | 34.20 | 26.40 | 56.09 | 15.84 | 3.00 | 43.66 | 11.48 | 10.67 | 46.01 | 46.74 | 53.50 |
| **Other** | | | | | | | | | | | | | | | | | | |
| SAM | ViT-L/16@1024 | 31.74 | 585.78 | 20.34 | 36.34 | 39.85 | 65.49 | 34.50 | 25.10 | 53.92 | 16.16 | 2.70 | 42.37 | 9.25 | 2.00 | 44.44 | 35.65 | 50.50 |
| SAM | ViT-H/16@1024 | 32.37 | 648.96 | 22.30 | 36.31 | 40.52 | 65.20 | 34.10 | 26.00 | 54.44 | 15.56 | 2.40 | 42.39 | 8.75 | 2.00 | 45.36 | 34.83 | 55.25 |
| MiDaS 3.0 | ViT-L/16@384 | 35.65 | 981.36 | 38.57 | 40.93 | 49.04 | 63.41 | 31.80 | 25.70 | 54.72 | 16.36 | 2.60 | 43.19 | 11.24 | 6.67 | 44.97 | 38.78 | 53.40 |
| MiDaS 3.1 | ViT-L/16@518 | 35.44 | 983.34 | 34.79 | 40.20 | 48.53 | 64.60 | 33.90 | 25.00 | 55.15 | 15.64 | 2.60 | 42.76 | 12.08 | 6.66 | 43.66 | 39.63 | 52.58 |
| Diffusion | SD2.1/16@512 | 36.59 | 1,044.28 | 37.71 | 42.00 | 48.38 | 64.55 | 33.40 | 25.70 | 56.99 | 15.56 | 3.10 | 43.14 | 10.40 | 9.33 | 45.88 | 44.68 | 52.40 |
| SupViT | ViT-L/16@224 | 40.13 | 1,197.39 | 46.55 | 54.72 | 57.27 | 65.94 | 34.00 | 28.00 | 56.22 | 16.44 | 3.10 | 43.52 | 11.82 | 16.67 | 46.67 | 48.49 | 52.75 |
| SupViT | ViT-H/14@224 | 37.45 | 1,082.43 | 42.61 | 48.45 | 52.98 | 63.51 | 35.29 | 26.50 | 55.78 | 15.16 | 3.30 | 44.16 | 11.49 | 4.66 | 43.79 | 44.55 | 52.91 |

Table 9: **All Benchmark Results for 0M Adapter Data + 737K Instruction Tuning Data**

**0.5M Adapter Data + 737K Instruction Finetune**   As shown in Table 10 and Table 9, the inclusion of an alignment stage with 0.5M data split results in a notable increase in performance for DFN-CLIP ViT-H/14@378, from 36.21 to 49.94. This substantial improvement highlights the value of the alignment stage for enhancing certain vision backbones, suggesting its importance in harnessing the full potential of vision models.

**1.2M Adapter Data + 737K Instruction Finetune**   As we increase the amount of data in the alignment phase, we observe a consistent performance improvement for SigLIP ViT-SO400M/14@384

| Vision Backbone | | General | | | | Knowledge | | | | OCR & Chart | | | | Vision-Centric | | | |
|---|---|---|---|---|---|---|---|---|---|---|---|---|---|---|---|---|---|
| Model | Architecture | Average | MME$^P$ | MMB | SEED$^I$ | GQA | SQA$^I$ | MMMU$^V$ | MathVista$^M$ | AI2D | ChartQA | OCRBench | TextVQA | DocVQA | MMVP | RealWorldQA | CV-Bench$^{2D}$ | CV-Bench$^{3D}$ |
| **Language-Supervised** | | | | | | | | | | | | | | | | | | |
| OpenAI CLIP | ViT-L/14@336 | 49.03 | 1,413.51 | 60.34 | 62.17 | 60.81 | 69.76 | 36.49 | 29.90 | 58.48 | 36.80 | 30.20 | 57.63 | 30.98 | 21.33 | 51.63 | 52.57 | 54.75 |
| DFN-CLIP | ViT-L/14@224 | 45.59 | 1,382.75 | 57.36 | 63.26 | 60.54 | 66.81 | 35.09 | 29.20 | 57.71 | 22.88 | 23.45 | 52.27 | 18.82 | 21.33 | 51.31 | 49.59 | 50.63 |
| DFN-CLIP | ViT-H/14@378 | 50.62 | 1,500.45 | 62.64 | 66.44 | 62.53 | 70.75 | 35.69 | 30.30 | 58.78 | 39.20 | 29.80 | 56.98 | 31.39 | 29.33 | 53.59 | 54.96 | 52.58 |
| EVA-CLIP-02 | ViT-L/14@336 | 47.13 | 1,362.07 | 62.64 | 63.96 | 61.66 | 69.46 | 35.89 | 27.90 | 56.96 | 20.96 | 26.10 | 53.93 | 19.07 | 20.00 | 53.07 | 56.28 | 58.16 |
| SigLIP | ViT-L/16@384 | 48.11 | 1,381.48 | 61.79 | 61.87 | 59.45 | 70.25 | 35.99 | 28.80 | 57.58 | 28.76 | 28.20 | 54.90 | 25.60 | 26.00 | 52.29 | 52.89 | 56.33 |
| SigLIP | ViT-SO400M/14@384 | 50.41 | 1,327.79 | 62.13 | 63.92 | 61.31 | 70.38 | 36.99 | 30.00 | 59.52 | 40.08 | 33.20 | 60.37 | 36.58 | 22.00 | 53.99 | 55.59 | 54.08 |
| OpenCLIP | ConvNeXt-L@512 | 48.01 | 1,366.85 | 59.66 | 62.89 | 61.31 | 68.77 | 36.99 | 28.50 | 59.29 | 27.88 | 25.50 | 57.57 | 29.48 | 16.00 | 53.20 | 53.81 | 55.91 |
| OpenCLIP | ConvNeXt-L@1024 | 40.29 | 1,084.62 | 12.94 | 51.02 | 49.78 | 65.47 | 34.20 | 27.60 | 56.36 | 29.92 | 13.25 | 50.37 | 43.67 | 13.33 | 49.08 | 51.85 | 41.58 |
| OpenCLIP | ConvNeXt-XXL@1024 | 50.45 | 1,405.65 | 57.96 | 63.58 | 62.41 | 68.02 | 34.30 | 29.40 | 59.62 | 42.96 | 26.20 | 61.82 | 42.67 | 28.67 | 55.16 | 49.92 | 54.16 |
| **Self-Supervised** | | | | | | | | | | | | | | | | | | |
| DINOv2 | ViT-L/14@336 | 42.64 | 1,283.95 | 54.64 | 59.03 | 60.19 | 66.39 | 35.29 | 25.70 | 58.03 | 16.00 | 3.20 | 45.39 | 11.79 | 20.00 | 50.59 | 53.51 | 58.33 |
| MoCo v3 | ViT-B/16@224 | 38.50 | 1,159.10 | 40.00 | 51.37 | 54.97 | 63.35 | 33.70 | 28.65 | 55.51 | 16.36 | 3.30 | 44.42 | 11.42 | 10.67 | 46.14 | 45.80 | 52.00 |
| MoCo v3 | ViT-L/16@224 | 37.71 | 1,074.13 | 41.19 | 49.46 | 53.61 | 63.66 | 33.70 | 27.40 | 55.83 | 17.04 | 3.40 | 43.84 | 11.98 | 8.00 | 46.27 | 48.69 | 45.59 |
| MAE | ViT-L/16@224 | 39.99 | 1,138.35 | 44.60 | 54.91 | 56.69 | 65.64 | 36.19 | 27.90 | 56.48 | 17.20 | 3.20 | 44.45 | 12.42 | 14.00 | 47.32 | 48.83 | 53.08 |
| I-JEPA | ViT-L/14@224 | 39.91 | 1,180.12 | 44.26 | 52.86 | 55.32 | 65.94 | 34.40 | 27.00 | 57.16 | 15.88 | 3.20 | 44.36 | 11.61 | 13.33 | 46.27 | 52.19 | 55.83 |
| **Other** | | | | | | | | | | | | | | | | | | |
| SAM | ViT-H/16@1024 | 32.18 | 649.99 | 22.47 | 36.37 | 40.46 | 64.60 | 32.50 | 25.80 | 54.66 | 15.80 | 2.70 | 42.40 | 8.89 | 0.00 | 45.62 | 37.02 | 53.08 |
| MiDaS 3.0 | ViT-L/16@384 | 40.07 | 1,183.95 | 47.40 | 53.00 | 56.15 | 66.19 | 32.90 | 27.60 | 56.61 | 17.00 | 3.00 | 44.34 | 11.55 | 19.33 | 47.32 | 45.01 | 54.08 |
| Diffusion | SD2.1/16@512 | 38.26 | 1,123.46 | 42.04 | 50.66 | 53.63 | 65.74 | 33.30 | 24.50 | 57.48 | 14.52 | 3.30 | 43.95 | 10.62 | 10.00 | 43.53 | 48.24 | 54.50 |
| SupViT | ViT-L/16@224 | 39.66 | 1,186.88 | 48.43 | 54.28 | 56.35 | 65.49 | 33.00 | 28.10 | 57.16 | 17.56 | 2.80 | 44.92 | 12.23 | 12.67 | 47.06 | 43.59 | 51.67 |

Table 10: **All Benchmark Results for 0.5M Adapter Data + 737K Instruction Tuning Data**

from 46.79 to 49.72 to 53.09 across 0M, 0.5M to 1.2M data splits as shown in Table 9, Table 10 and Table 11.

| Vision Backbone | | General | | | | Knowledge | | | | OCR & Chart | | | | Vision-Centric | | | |
|---|---|---|---|---|---|---|---|---|---|---|---|---|---|---|---|---|---|
| Model | Architecture | Average | MME$^P$ | MMB | SEED$^I$ | GQA | SQA$^I$ | MMMU$^V$ | MathVista$^M$ | AI2D | ChartQA | OCRBench | TextVQA | DocVQA | MMVP | RealWorldQA | CV-Bench$^{2D}$ | CV-Bench$^{3D}$ |
| **Language-Supervised** | | | | | | | | | | | | | | | | | | |
| OpenAI CLIP | ViT-L/14@336 | 50.49 | 1,476.65 | 61.96 | 65.45 | 62.78 | 69.06 | 35.00 | 29.50 | 58.94 | 37.84 | 30.90 | 58.21 | 32.11 | 28.66 | 54.90 | 54.14 | 54.60 |
| DFN-CLIP | ViT-L/14@224 | 46.01 | 1,341.14 | 56.68 | 63.74 | 60.75 | 66.96 | 33.80 | 28.65 | 57.04 | 23.32 | 23.20 | 52.85 | 18.97 | 26.67 | 51.44 | 52.91 | 52.08 |
| DFN-CLIP | ViT-H/14@378 | 51.17 | 1,426.32 | 62.38 | 67.29 | 62.89 | 69.01 | 35.89 | 30.00 | 60.01 | 41.08 | 30.60 | 57.53 | 31.69 | 32.67 | 55.95 | 55.46 | 55.00 |
| EVA-CLIP-02 | ViT-L/14@336 | 49.71 | 1,449.78 | 64.00 | 67.53 | 63.60 | 69.91 | 35.49 | 28.40 | 59.16 | 24.76 | 27.10 | 55.39 | 21.63 | 34.67 | 55.69 | 57.83 | 57.75 |
| SigLIP | ViT-L/16@384 | 50.87 | 1,424.20 | 59.40 | 65.48 | 62.56 | 68.67 | 35.99 | 29.70 | 59.32 | 43.76 | 33.50 | 59.59 | 35.20 | 28.00 | 53.33 | 55.42 | 56.08 |
| SigLIP | ViT-SO400M/14@384 | 53.91 | 1,455.64 | 63.66 | 67.62 | 63.70 | 72.10 | 36.09 | 29.30 | 61.59 | 43.76 | 37.20 | 61.82 | 40.19 | 36.60 | 56.99 | 59.61 | 59.58 |
| OpenCLIP | ConvNeXt-L@512 | 49.16 | 1,416.87 | 60.60 | 63.87 | 61.87 | 69.92 | 35.79 | 29.50 | 59.36 | 34.40 | 28.00 | 58.36 | 28.41 | 27.33 | 51.90 | 54.64 | 51.80 |
| OpenCLIP | ConvNeXt-L@1024 | 51.00 | 1,392.92 | 58.21 | 65.47 | 62.89 | 67.43 | 34.90 | 29.90 | 59.13 | 30.44 | 13.50 | 51.47 | 44.13 | 26.67 | 55.29 | 53.57 | 55.08 |
| OpenCLIP | ConvNeXt-XXL@1024 | 52.18 | 1,402.94 | 59.40 | 65.21 | 62.73 | 68.27 | 33.10 | 29.30 | 59.84 | 48.00 | 28.00 | 63.27 | 48.11 | 34.67 | 55.95 | 53.83 | 55.08 |
| **Self-Supervsied** | | | | | | | | | | | | | | | | | | |
| DINOv2 | ViT-L/14@336 | 41.85 | 1,190.81 | 51.83 | 56.90 | 60.38 | 66.04 | 34.20 | 27.40 | 56.41 | 16.44 | 3.30 | 45.12 | 11.79 | 21.33 | 49.67 | 53.91 | 55.33 |
| MoCo v3 | ViT-B/16@224 | 38.88 | 1,129.32 | 41.62 | 52.19 | 55.03 | 65.89 | 33.30 | 28.30 | 56.44 | 16.48 | 3.00 | 44.09 | 11.47 | 12.00 | 47.58 | 45.17 | 53.00 |
| MoCo v3 | ViT-L/16@224 | 37.07 | 1015.20 | 37.28 | 48.31 | 52.63 | 65.49 | 34.50 | 27.60 | 55.41 | 16.92 | 3.10 | 43.57 | 11.50 | 14.67 | 45.49 | 45.17 | 40.75 |
| MAE | ViT-L/16@224 | 40.39 | 1,132.80 | 43.40 | 55.67 | 57.42 | 66.04 | 35.59 | 27.60 | 56.48 | 17.36 | 3.30 | 44.53 | 12.30 | 16.00 | 47.71 | 49.24 | 56.55 |
| I-JEPA | ViT-H/14@224 | 40.27 | 1,207.88 | 45.79 | 54.51 | 56.15 | 65.29 | 34.40 | 27.10 | 56.19 | 16.20 | 3.20 | 43.45 | 11.58 | 18.00 | 45.88 | 49.57 | 56.58 |
| **Other** | | | | | | | | | | | | | | | | | | |
| SAM | ViT-H/16@1024 | 32.54 | 682.81 | 23.32 | 36.16 | 40.32 | 65.20 | 33.20 | 26.50 | 54.21 | 15.68 | 2.50 | 41.76 | 8.98 | 1.33 | 46.80 | 37.66 | 52.90 |
| MiDaS 3.0 | ViT-L/16@384 | 39.15 | 1,132.18 | 46.21 | 51.75 | 55.57 | 66.30 | 33.70 | 26.70 | 56.06 | 17.08 | 3.10 | 43.65 | 11.66 | 15.30 | 45.75 | 44.44 | 52.58 |
| Diffusion | SD2.1/16@512 | 39.51 | 1,168.52 | 40.00 | 53.80 | 55.33 | 64.60 | 35.00 | 26.10 | 57.16 | 15.36 | 3.10 | 44.23 | 11.06 | 18.67 | 47.32 | 48.04 | 53.90 |
| Supervised | ViT-L/16@224 | 39.12 | 1,216.11 | 45.28 | 51.46 | 55.88 | 64.15 | 34.70 | 26.80 | 55.76 | 16.80 | 2.80 | 44.42 | 11.61 | 11.33 | 47.97 | 44.62 | 51.60 |

Table 11: **All Benchmark Results for 1.2M Adapter Data + 737K Instruction Tuning Data**

**1.2M Adapter Data + 737K Instruction Finetune with *Unfrozen* Vision Model**   Here, we present the results of different vision models trained with 1.2m adapter data and 737K instruction tuning data in Appendix F.2. Comparing to Appendix I.2, we observe nearly all the models see improvement on most of the benchmarks, especially on the OCR & Chart and Vision-Centric benchmarks.

**1.2M Adapter Data + 5M Instruction Finetune**   We present the results of 5M instruction tuning experiments in Fig. 6 here. In Table 12, we observe that after 5m instruction tuning, the gap between DINOv2 and CLIP models continue to bridge on general, knowledge and vision-centric benchmarks.

## F.3   Model Ensemble

**Model Ensemble Details**   We introduce the implementation details of the model ensemble in Section 2.5. For a given image, the image passes through each vision encoder to obtain the features

| Model | Architecture | Average | MME$^P$ | MMB | SEED$^I$ | GQA | SQA$^I$ | MMMU$^V$ | MathVista$^M$ | AI2D | ChartQA | OCRBench | TextVQA | DocVQA | MMVP | RealWorldQA | CV-Bench$^{2D}$ | CV-Bench$^{3D}$ |
|---|---|---|---|---|---|---|---|---|---|---|---|---|---|---|---|---|---|---|
| Other | | | | | | | | | | | | | | | | | | |
| OpenAI CLIP | ViT-L/14@336 | 52.90 | 1,477.15 | 63.15 | 68.49 | 64.24 | 69.21 | 34.90 | 28.70 | 61.24 | 47.68 | 40.20 | 58.92 | 35.22 | 26.00 | 58.82 | 59.65 | 56.16 |
| DFN-CLIP | ViT-L/14@224 | 45.80 | 1,364.91 | 55.15 | 63.06 | 59.94 | 67.28 | 35.59 | 29.80 | 58.65 | 20.48 | 23.40 | 52.23 | 18.29 | 22.67 | 50.85 | 53.59 | 53.50 |
| EVA-CLIP-02 | ViT-L/14@336 | 51.30 | 1,492.35 | 65.53 | 69.75 | 65.18 | 68.62 | 35.00 | 29.50 | 60.78 | 29.32 | 29.90 | 56.87 | 21.59 | 44.67 | 58.95 | 54.66 | 55.91 |
| SigLIP | ViT-L/16@384 | 52.47 | 1,429.11 | 63.57 | 67.34 | 63.44 | 68.02 | 36.09 | 29.70 | 61.56 | 46.68 | 35.70 | 59.86 | 35.93 | 32.67 | 55.29 | 55.24 | 57.00 |
| SigLIP | ViT-SO400M/14@384 | 55.27 | 1,489.05 | 66.55 | 69.59 | 64.58 | 70.45 | 35.69 | 29.20 | 62.34 | 51.28 | 40.80 | 63.28 | 43.02 | 38.00 | 59.61 | 61.58 | 53.91 |
| OpenCLIP | ConvNeXt-L@512 | 52.76 | 1,467.38 | 63.40 | 66.92 | 63.17 | 69.16 | 34.90 | 29.70 | 58.45 | 52.04 | 35.40 | 61.87 | 38.79 | 30.67 | 56.21 | 54.24 | 55.91 |
| Other | | | | | | | | | | | | | | | | | | |
| DINOv2 | ViT-L/14@336 | 43.26 | 1,261.43 | 53.96 | 63.22 | 62.61 | 65.49 | 34.50 | 27.70 | 56.90 | 15.40 | 3.40 | 44.87 | 11.22 | 26.00 | 54.38 | 53.06 | 56.40 |
| MoCo v3 | ViT-B/16@224 | 39.51 | 1,175.34 | 41.70 | 53.31 | 56.15 | 65.25 | 33.40 | 28.30 | 55.76 | 15.48 | 3.20 | 44.42 | 11.10 | 18.00 | 46.67 | 45.38 | 55.25 |
| MoCo v3 | ViT-L/16@224 | 37.59 | 1075.39 | 39.15 | 50.14 | 53.65 | 65.49 | 34.60 | 27.30 | 55.44 | 17.28 | 3.00 | 44.21 | 11.70 | 14.67 | 44.58 | 45.38 | 41.12 |
| MAE | ViT-L/16@224 | 41.43 | 1,181.51 | 45.53 | 58.92 | 58.75 | 64.65 | 35.00 | 29.20 | 57.12 | 16.88 | 3.10 | 44.67 | 11.74 | 18.67 | 49.67 | 53.04 | 56.83 |
| I-JEPA | ViT-H/14@224 | 41.90 | 1,175.70 | 48.00 | 59.60 | 59.35 | 64.45 | 35.09 | 27.60 | 57.32 | 16.20 | 3.00 | 45.50 | 11.40 | 22.67 | 49.93 | 52.38 | 59.08 |
| Other | | | | | | | | | | | | | | | | | | |
| MiDaS 3.0 | ViT-L/16@384 | 38.28 | 1,065.26 | 42.64 | 50.95 | 56.10 | 65.39 | 35.00 | 27.20 | 52.98 | 15.96 | 2.80 | 43.49 | 11.23 | 12.00 | 46.14 | 43.41 | 53.90 |
| Supervised | ViT-L/16@224 | 40.01 | 1,222.41 | 47.40 | 54.15 | 57.26 | 64.35 | 34.40 | 26.40 | 56.09 | 16.20 | 3.20 | 44.73 | 11.74 | 14.00 | 46.41 | 49.33 | 53.40 |

Table 12: **All Benchmark Results for 1.2M Adapter Data + 737K Instruction Tuning Data with *Unfrozen* vision model**.

| Method | Backbone | Unfreeze | Average | MME$^P$ | MMB | SEED$^I$ | GQA | SQA$^I$ | MMMU$^V$ | MathVista$^M$ | AI2D | ChartQA | OCRBench | TextVQA | DocVQA | MMVP | RealWorldQA | CV-Bench$^{2D}$ | CV-Bench$^{3D}$ |
|---|---|---|---|---|---|---|---|---|---|---|---|---|---|---|---|---|---|---|---|
| OpenAI CLIP | ViT-L/14@336 | × | 55.85 | 1,577.33 | 69.70 | 70.22 | 63.33 | 73.67 | 36.19 | 36.60 | 64.80 | 49.12 | 36.90 | 60.33 | 39.79 | 32.67 | 55.56 | 66.11 | 59.75 |
| DINOv2 | ViT-L/14@336 | × | 45.36 | 1,373.14 | 57.02 | 64.58 | 61.67 | 67.13 | 36.19 | 30.70 | 60.62 | 19.04 | 3.40 | 46.39 | 13.27 | 26.67 | 52.68 | 59.81 | 57.91 |
| OpenAI CLIP | ViT-L/14@336 | ✓ | 57.44 | 1,585.34 | 68.68 | 71.47 | 63.96 | 77.39 | 36.09 | 37.30 | 65.12 | 59.36 | 48.00 | 62.39 | 45.24 | 31.33 | 56.21 | 61.09 | 56.16 |
| DINOv2 | ViT-L/14@336 | ✓ | 47.40 | 1,366.65 | 61.62 | 69.72 | 63.68 | 68.72 | 36.29 | 35.50 | 60.88 | 18.64 | 4.40 | 47.92 | 14.66 | 34.67 | 54.64 | 60.98 | 57.83 |

Table 13: **All Benchmark Results for 1.2M Adapter Data + 5M Instruction Tuning Data**

from the last layer. The shape of each model's output differs depending on the resolution and patch size of each vision model. To resolve these differences, we interpolate the output of each model to a fixed number of tokens, using 576 tokens in our implementation, as described in Section 2.5. Our example code for interpolation can be seen below.

```
# Example code for interpolation
b, num_tokens, dim = image_features.shape
if num_tokens != self.image_token_len:
    target_h = target_w = int(np.sqrt(self.image_token_len))
    h = w = int(np.sqrt(num_tokens))
    image_features = image_features.view(b, h, w, dim)
    image_features = image_features.permute(0, 3, 1, 2).contiguous()
    image_features = F.interpolate(image_features), size=(target_h, target_w), mode='bilinear', align_corners=False)
    image_features = image_features.permute(0, 2, 3, 1).contiguous().flatten(1, 2)
```

We then concatenate the model outputs along the feature dimension and use a larger MLP to project the concatenated visual tokens into the LLM token space.

**Full results on Model Ensemble**   We present all the benchmarks from the model ensemble experiment in Section 2.5 in Table 14. As discussed in Section 1 and Section 2.4, this comprehensive view of benchmarks provides a better understanding of the model's performance compared to simply averaging across benchmarks. Adding a vision-only SSL model enhances the MLLM's performance in vision-centric benchmarks while maintaining strong capabilities in other categories.

# G   Data

## G.1   Catalog of Visual Instruction Data

Here, we provide a comprehensive catalog of visual instruction datasets utilized in our study. The datasets are categorized based on their primary focus, including general conversation and VQA data, OCR-related data, counting data, knowledge-based data, and language-only data. Table 15 summarizes these datasets and their respective references.

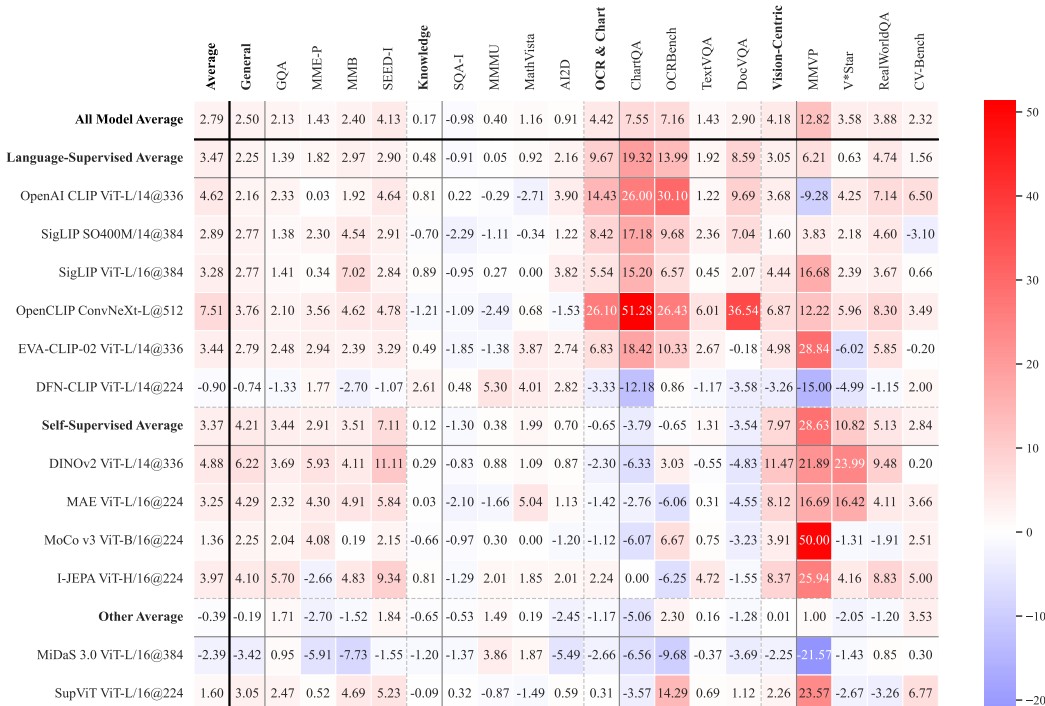

Figure 13: **Percentage (%) Change in Benchmark Performance (Frozen → Unfrozen)**
Heatmap depicting the percentage change in performance across multiple benchmarks when visual encoders are unfrozen compared to when they are kept frozen during fine-tuning. The color gradient indicates the magnitude of the performance change after unfreezing visual encoders—white indicates no change, red is a positive change, and blue is a negative change. Notably, unfreezing leads to significant gains in OCR Chart tasks for most Language-Supervised Models, as reflected by the deep red cells. ConvNeXt, in particular, shows substantial improvements, demonstrating the benefits of updating this visual encoder during fine-tuning.

| Method | Average | General | | | | Knowledge | | | | OCR & Chart | | | | Vision-Centric | | | |
|---|---|---|---|---|---|---|---|---|---|---|---|---|---|---|---|---|---|
| | | $MME^P$ | MMB | $SEED^I$ | GQA | $SQA^I$ | $MMMU^V$ | $MathVista^M$ | AI2D | ChartQA | OCRBench | TextVQA | DocVQA | MMVP | RealWorldQA | $CV\text{-}Bench^{2D}$ | $CV\text{-}Bench^{3D}$ |
| SigLIP+DINOv2 | 51.61 | 1,432.02 | 61.28 | 65.99 | 63.30 | 68.82 | 35.69 | 29.40 | 60.01 | 43.00 | 35.70 | 60.40 | 37.54 | 30.00 | 53.99 | 55.52 | 53.58 |
| SigLIP+DINOv2+ConvNext | 54.52 | 1,503.51 | 63.83 | 67.97 | 63.95 | 70.40 | 35.99 | 29.30 | 60.69 | 48.20 | 36.90 | 64.97 | 45.53 | 34.67 | 58.69 | 55.74 | 60.33 |
| SigLIP+DINOv2+ConvNext+CLIP | 54.74 | 1,479.46 | 63.32 | 67.63 | 64.04 | 71.39 | 35.49 | 29.10 | 59.88 | 50.24 | 39.60 | 64.55 | 46.12 | 32.67 | 58.95 | 58.54 | 60.42 |
| SigLIP+ConvNext | 54.53 | 1,494.97 | 64.60 | 67.98 | 63.58 | 71.05 | 34.90 | 29.80 | 60.85 | 50.64 | 38.00 | 64.53 | 46.52 | 32.00 | 57.91 | 58.83 | 56.58 |
| CLIP+ConvNext | 54.45 | 1,511.08 | 63.83 | 67.41 | 63.63 | 70.80 | 35.09 | 30.40 | 59.91 | 51.32 | 35.00 | 64.45 | 47.88 | 33.33 | 57.25 | 56.32 | 59.08 |
| SigLIP+DINOv2+ConvNext-L | 53.78 | 1,450.64 | 63.57 | 67.79 | 63.63 | 71.34 | 34.80 | 30.20 | 61.04 | 49.32 | 37.70 | 64.05 | 45.83 | 30.00 | 56.21 | 58.08 | 54.33 |
| SigLIP+CLIP+ConvNext-L | 54.53 | 1,507.28 | 63.23 | 68.64 | 63.63 | 71.10 | 35.89 | 30.90 | 59.97 | 52.36 | 38.50 | 65.40 | 47.92 | 28.67 | 57.25 | 57.66 | 55.92 |

Table 14: **All Benchmark Results for Model Ensemble with 1.2M Adapter Data + 737K Instruction Tuning Data.** Here, "SigLIP" = ViT-SO400M/14@384, "DINOv2" = ViT-L/14@518, "ConvNext" = OpenCLIP ConvNeXt-XXL@1024, and "CLIP" = OpenAI CLIP ViT-L/14@336.

## G.2 Additional System Prompts used in Cambrian Data

Here, we investigate a phenomenon we term the "answer machine phenomenon". We observe that a well-trained MLLM may excel at VQA benchmarks, but lack basic conversational abilities and default to outputting short, curt responses (see examples in Fig. 14). This discrepancy arises because benchmark questions typically require responses that are limited to a single option, choice, or word—diverging from the more broad and realistic use cases of MLLMs. Similar phenomena have been discussed in other LLM studies [114, 151, 155].

| Category | Datasets |
|---|---|
| **General Conversation & VQA Data** | LVIS-Instruct4V [131], SketchyVQA [129], OODVQA [129], VizWiz [51], ALLaVA [25], IDK [22], Q-Instruct [135], LAION GPT-4V [70], Hateful-Memes [66], Visual7W [159], Visualmrc [123], AlfWorld [117], LNQA [108], LLaVA150K [82], ShareGPT [27], VQAv2 [50], GQA [57], OKVQA [95], A-OKVQA [116], RefCOCO [140], VisualGenome [69], GPT-4V recorded chat |
| **OCR Related Data** | LLAVAR [149], ChartQA [96], DocVQA [97], DVQA [61], ArxivQA [76], AI2D [65], ScreenQA [56], SynthDog [67], IconQA [89], WTQ [106], WikiSQL [153], FinQA [31], HiTab [32], TAT-QA [158], TabMWP [88], Chart2Text [62], VisText [124], InfoVQA [16], ST-VQA [17], Rendered-Text [134], OCRVQA [100], TextCaps [119], ShareGPTOCRData [27] |
| **Counting Data** | TallyQA [1], CLEVR [59] |
| **Knowledge-Based Data** | **Code:** Design2Code [118], WebSight [71], Datikz [14]
**Math:** MathVision [132], Geo170K [42], TQA [5], Inter-GPS [90], RAVEN [148], GeomVerse [64]
**Science:** ScienceQA [91], PathVQA [53] |
| **Language Only Data** | Dolly [36], MathInstruct [142], WizardCoder [93], OrcaMath [99], OpenCodeInterpreter [152], OpenOrca [78] |

Table 15: Visual Instruction-Tuning Data Catalog

We suspect that this issue stems from instruction tuning data containing an excessive number of short-response VQA tasks, leading to catastrophic forgetting in LLMs. To address this, we incorporate additional system prompts during training. We append prompts such as "*Answer the question using a single word or phrase.*" before questions that generate a single word or phrase in the response. Full details of the system prompts used are provided in Appendix G.2. After integrating these system prompts, we observe that while the model's benchmark performance remains unchanged, its conversational ability improves dramatically. For example, in Fig. 14, models with system prompts produce longer and more engaging responses while answering questions correctly. The system prompts also enhance the model's performance on reasoning-related tasks, such as math problems, by encouraging a chain of thoughts [133] followed by the answer.

This underscores the necessity of developing evaluation protocols like the Chatbot Arena [34] for MLLMs, despite the challenges in collecting large-scale, real-world interaction data. While performing well on benchmarks is important, it is equally crucial to ensure the model can engage in meaningful and natural interactions. The overall user experience and the model's conversational abilities are paramount, as a model that excels in benchmarks but fails to converse effectively cannot meet the needs of practical applications.

As our Cambrian data includes instructions/questions and responses of different types and formats (e.g., Short response with a single word or regular response as a complete sentence), it is important to specify the required response format in the instruction prompt to avoid ambiguity and possible conflicts. Some of the datasets already include such prompts and we add proper prompts for the remaining datasets. The detailed response formatting prompts we additionally add are listed in Table 16.

### G.3 Data Engine

**Comprehensive Implementation Details of the Data Engine**

The data engine is designed to generate instruction tuning data for knowledge-based fields, where previous works rarely covers and MLLMs are not reliable to distill for from. The data engine takes in a given field, such as "Physics", utilizing reliable web sources like Wikipedia. Below are the various stages involved in the process. We also visualize this process in Fig. 15:

**Stage 1 - Topic Generation:** We start by compiling a list of fields and subfields and subsequently generate topics for each field using a Large Language Model (LLM), such as GPT-4. In this stage,

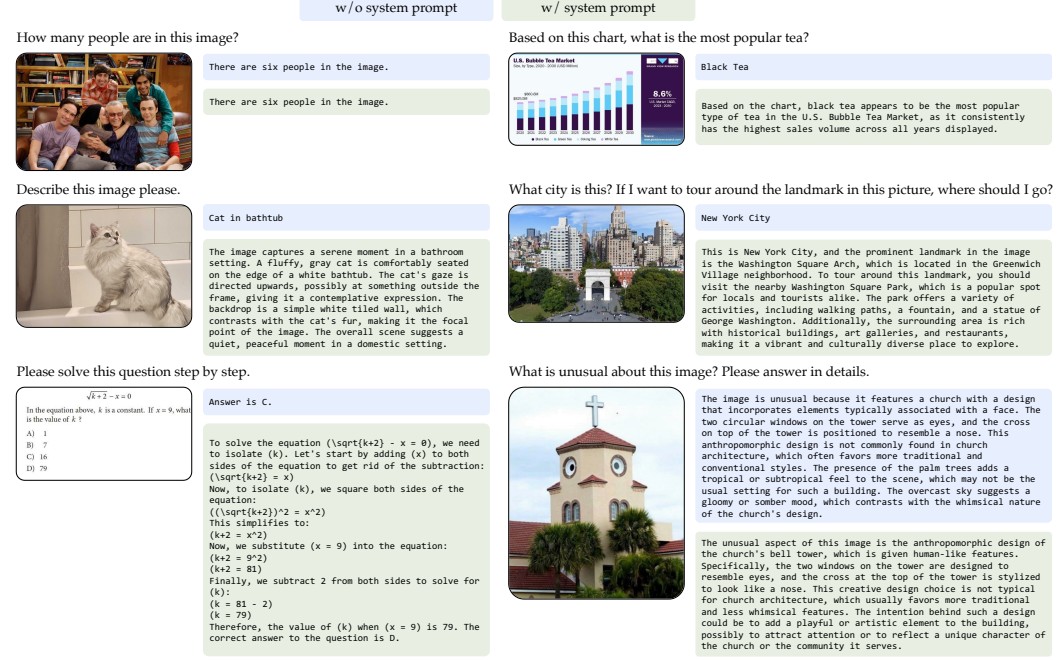

**Figure 14: Incorporating System Prompt in Instruction Tuning Data alleviates the "Answer Machine Phenomenon"** By adding system prompts in Cambrian-7M, the model exhibits better chat ability while retaining strong question answering abilities. The model without system prompts requires additional prompting to elicit longer responses.

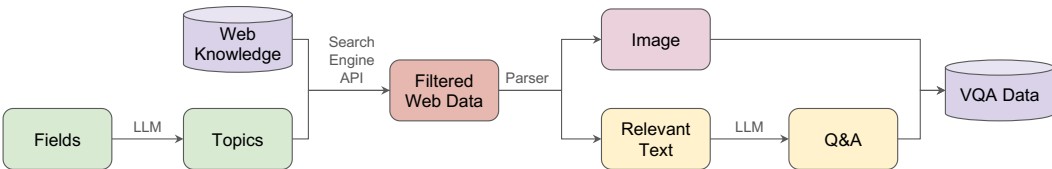

**Figure 15: Targeted Internet Data Collection Engine.** We build a targeted internet data engine to collect high-quality and large-scale multimodal instruction tuning data for domains like knowledge.

we processed 30 fields, resulting in 3660 topics. We then post-process the output of LLMs into json formats. For example, the topic data for Physics looks like below.

```
Physics
{
    "Classical Mechanics": [
        "Newton's Laws of Motion",
        "Conservation of Energy",
        "Conservation of Momentum",
        "Harmonic Motion",
        "Rotational Dynamics",
        "Gravitation and Orbits",
        "Fluid Dynamics",
        "Elasticity and Plasticity",
        "Friction",
        "Waves and Sound",
        "Velocity and Acceleration",
        "Angular Momentum",
        "Statics and Equilibrium",
        "Kinematics of Particles",
        "Dynamics of Systems of Particles",
        "Collisions",
        "Centripetal Force and Acceleration",
        "Lagrangian and Hamiltonian Mechanics",
        "Chaos Theory",
        "Equations of Motion"
    ],
    "Electromagnetism": [
        "Coulomb's Law",
        "Electric Field and Electric Potential",
        "Gauss's Law",
        "Capacitance and Dielectrics",
        "Current and Resistance",
```

| Index | Response formatting prompts |
|---|---|
| 1 | Answer the question using a single word or phrase. |
| 2 | Answer the question using a single number or phrase. |
| 3 | Answer with the option's letter from the given choices directly. |
| 4 | Give the short answer directly. |
| 5 | Answer the question using a single word or phrase. |
| 6 | When the provided information is insufficient, respond with <no answer>. |
| 7 | Directly provide the HTML code. |
| 8 | First show your reasoning process and then give the final answer. |
| 9 | When the provided information is insufficient, respond with 'Unanswerable'. Answer the question using a single word or phrase. |
| 10 | Answer with the letter. |

| Dataset | Prompts added |
|---|---|
| SketchyVQA | 1 |
| OODVQA | 1 |
| VizWiz | 9 |
| Q-Instruct | 1, 3 |
| ChartQA | 2 |
| DocVQA | 4 |
| DVQA | 1 |
| AI2D | 1 |
| ScreenQA | 1, 6 |
| CLEVR | 1 |
| TallyQA | 1 |
| PathVQA | 1 |
| MathInstruct | 8 |
| Design2Code | 7 |
| IconQA | 1, 10 |
| HiTab | 1 |
| WTQ | 1 |
| WikiSQL | 1 |
| Inter-GPS | 10 |
| Visual7W | 3 |
| TQA | 10 |
| RAVEN | 1 |

Table 16: Response formatting prompts for Cambrian Data

```
        "Direct Current Circuits",
        "Magnetic Fields and Magnetic Forces",
        "Ampere's Law",
        "Faraday's Law of Induction",
        "Inductance",
        "Alternating Current Circuits",
        "Electromagnetic Waves",
        "Maxwell's Equations",
        "Electromagnetic Radiation",
        "Optics and Light",
        "Quantum Electrodynamics",
        "Special Theory of Relativity Implication",
        "Magnetostatics",
        "Electrostatics",
        "Bioelectromagnetism"
    ],
    ...
}
```

**Stage 2 - Filtering Web Data:** For each generated topic, we utilize search engine APIs to fetch relevant high-quality web pages. For each topic, we query for 10 relevant links. Thus, we get 36,600 webpages post this stage. Here is an example of the data retrieved for the topic "Electric Field and Electric Potential":

```
"Electric Field and Electric Potential": [
    "https://en.wikipedia.org/wiki/Electric_potential",
    "https://en.wikipedia.org/wiki/Electric_field",
    "https://en.wikipedia.org/wiki/Electric_potential_energy",
```

```
    "https://en.wikipedia.org/wiki/Voltage",
    "https://en.wikipedia.org/wiki/Electricity",
    "https://en.wikipedia.org/wiki/Electrostatics",
    "https://en.wikipedia.org/wiki/Electric_dipole_moment",
    "https://en.wikipedia.org/wiki/Magnetic_vector_potential",
    "https://en.wikipedia.org/wiki/Electric-field_screening",
    "https://en.wikipedia.org/wiki/Electric_flux"
],
```

**Stage 3 - Parsing:** In this stage, we parse each web page to extract image-caption-text tuples. We aim to identify the blocks containing an image, the image's caption, and relevant textual content. Below is an example of the parsed data for the same topic, "Electric Field and Electric Potential":

```
{
    "Electric Field and Electric Potential",
    [
      {
        "section":  "Electrostatics",
        "text":  "An electric potential at a point r in a static electric field E is given by the line integral where C is an
arbitrary path from some fixed reference point to r; it is uniquely determined up to a constant...  The generalization of
electric potential to this case is described in the section Generalization to electrodynamics.",
        "images":  [
          {
            "url":  https://upload.wikimedia.org/wikipedia/commons/thumb/1/1e/VFPt_plus_thumb_potential+contour.svg/
142px-VFPt_plus_thumb_potential+contour.svg.png,
            "caption":  "Electric potential of separate positive and negative point charges shown as color range from magenta
(+), through yellow (0), to cyan (-).  Circular contours are equipotential lines.  Electric field lines leave the positive
charge and enter the negative charge."
          },
          {
            "url":  https://upload.wikimedia.org/wikipedia/commons/thumb/e/e0/VFPt_charges_plus_minus_potential+contour.svg/
288px-VFPt_charges_plus_minus_potential+contour.svg.png,
            "caption":  "Electric potential in the vicinity of two opposite point charges."
          }
        ],
        "link":  https://en.wikipedia.org/wiki/Electric_potential,
        "title":  "Electric potential",
        "field":  "Physics",
        "subfield":  "Electromagnetism",
        "topic":  "Electric Field and Electric Potential"
      },
      ...
    ]
}
```

**Stage 4 - Data Generation:** We generate dataset in this stage, ensuring high quality. We first filter out data samples with *fewer than 50 words* in the text. Then, instead of downloading images directly from the links retrieved during web parsing, we download high-resolution images from the original sources. We then convert formats like SVG or GIF into a common standardized format, PNG.

Question-Answer pairs are generated by using LLM such as GPT-3.5 from the image metadata, caption, and contextual text. These Q&A pairs and the image form our VQA dataset. We generated 165k data samples. Here is an example of the generated data:

```
{
    "id":  "96232.png",
    "image_id":  "450px-Magnetic_Vector_Potential_Circular_Toroid",
    "image_url":  "...",
    "text":  "...",
    "caption":  "Representing the Coulomb gauge magnetic vector potential A, magnetic
flux density B, and current density J fields around a toroidal inductor of circular
cross section.  Thicker lines indicate field lines of higher average intensity.
Circles in the cross section of the core represent the B-field coming out of the
picture, plus signs represent B-field going into the picture.  \u2207 \u22c5 A = 0
```

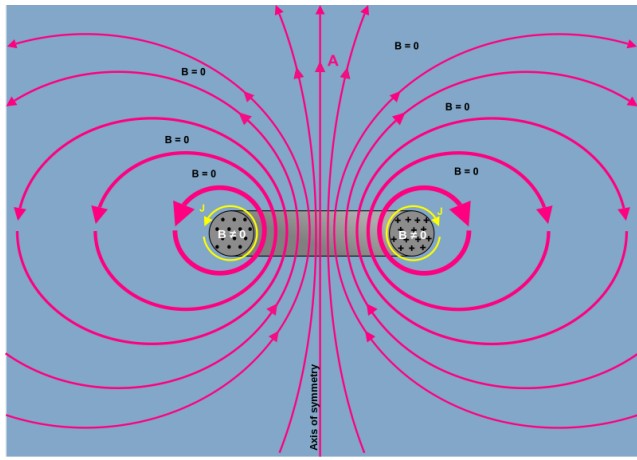

Figure 16: **Dataset Image Id:** `96232.png`

| | | General | | | | Knowledge | | | | OCR & Chart | | | | Vision-Centric | | | |
|---|---|---|---|---|---|---|---|---|---|---|---|---|---|---|---|---|---|
| # of Data | Average | MME$^P$ | MMB | SEED$^I$ | GQA | SQA$^I$ | MMMU$^V$ | MathVista$^M$ | AI2D | ChartQA | OCRBench | TextVQA | DocVQA | MMVP | RealWorldQA | CV-Bench$^{2D}$ | CV-Bench$^{3D}$ |
| $t = 150k$  4015k | 53.74 | 1,512.3 | 67.0 | 68.3 | 61.2 | 73.4 | 35.1 | 34.3 | 62.4 | 44.6 | 39.1 | 58.5 | 38.5 | 30.0 | 55.2 | 61.98 | 54.7 |
| $t = 250k$  5218k | 54.31 | 1,475.9 | 67.3 | 69.2 | 61.6 | 73.4 | 35.9 | 34.5 | 62.4 | 46.5 | 36.9 | 59.2 | 38.6 | 32.0 | 56.6 | 63.68 | 57.5 |
| $t = 350k$  5883k | 54.27 | 1,461.9 | 66.2 | 68.9 | 61.6 | 73.8 | 36.4 | 32.8 | 62.5 | 46.8 | 38.3 | 59.3 | 39.3 | 31.3 | 54.9 | 62.68 | 60.4 |
| $t = 450k$  6383k | 54.15 | 1,534.1 | 67.6 | 66.3 | 61.9 | 72.9 | 35.1 | 36.9 | 63.8 | 45.8 | 38.6 | 58.4 | 39.4 | 28.0 | 53.6 | 64.60 | 56.8 |

Table 17: **All Benchmark Results for Data Balancing Experiments**

```
has been assumed",
   "section":  "Depiction of the A-field",
   "...",
   "Question":  "What do the thicker lines signify?",
   "Answer":  "The thicker lines in the image represent paths of higher average
intensity in the magnetic vector potential A field.  Shorter paths have higher
intensity, ensuring that the path integral remains constant.  These lines are drawn
to visually convey the general behavior and intensity distribution of the A-field
around the toroidal inductor"
}
```

This data engine is designed for scalability and efficiency and is capable of handling extensive data generation tasks using multithreading techniques.

### G.4   Full results on data curation experiment

**Data Balance via Fitlering** $t$ As discussed in Section 4.2, if left unfiltered, the data pool is dominated by noisy, unbalanced data sources such as CLEVR and DVQA, leading to pronounced exponential tails. However, as we apply different $t$ values to filter data from each source, the exponential tails become less pronounced, resulting in a more balanced dataset. We also present all the results in Table 17. $t$ value 250k has the highest average across all benchmarks; 250k and 350k also have the highest performance across many individual benchmarks.

Here, we plot the cumulative sum of counts for entries sorted by counts from tail to head. From Fig. 17, we see this intermediate threshold prevents explosive heavy tail.

**Data Ratio Studies** We present the full results of our data ratio study in Table 18. The table highlights the importance of finding an optimal data ratio that balances different aspects of MLLM. Experiment 5 achieves well-rounded performance with its selected data ratio.

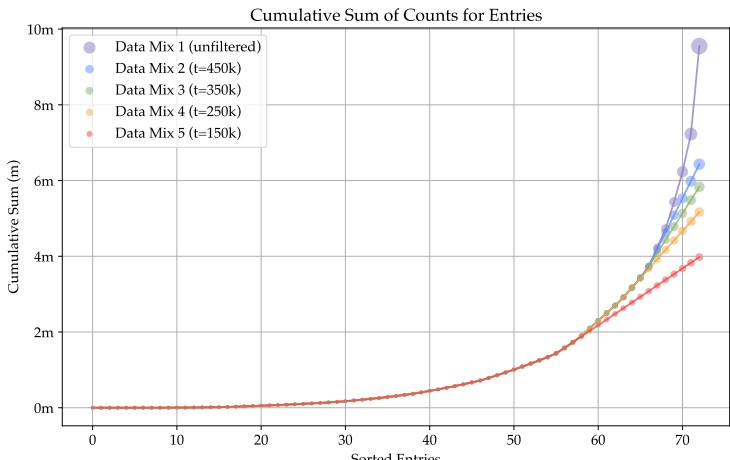

Figure 17: **Data Balancing via Applying Thresholds on Data Sources**. Applying threshold $t$ alleviates the exponential tail of Cambrian-10M.

| | | General | | | | Knowledge | | | | OCR & Chart | | | | Vision-Centric | | | |
|---|---|---|---|---|---|---|---|---|---|---|---|---|---|---|---|---|---|
| | Average | MME$^P$ | MMB | SEED$^I$ | GQA | SQA$^I$ | MMMU$^V$ | MathVista$^M$ | AI2D | ChartQA | OCRBench | TextVQA | DocVQA | MMVP | RealworldQA | CV-Bench$^{2D}$ | CV-Bench$^{3D}$ |
| exp1 | 47.49 | 1,309.10 | 58.00 | 60.10 | 54.00 | 72.40 | 34.80 | 31.20 | 59.10 | 34.20 | 34.50 | 54.20 | 33.00 | 13.30 | 47.60 | 57.40 | 50.58 |
| exp2 | 47.78 | 1,351.70 | 60.30 | 61.20 | 55.40 | 72.80 | 35.20 | 29.50 | 59.10 | 31.20 | 33.40 | 54.20 | 30.50 | 15.30 | 48.20 | 58.40 | 52.25 |
| exp3 | 48.28 | 1,299.53 | 60.56 | 61.79 | 55.74 | 72.04 | 34.90 | 32.10 | 59.40 | 33.20 | 33.90 | 54.15 | 31.90 | 21.30 | 48.60 | 58.52 | 49.41 |
| exp4 | 47.47 | 1,288.98 | 58.16 | 61.47 | 55.00 | 71.05 | 37.10 | 28.20 | 58.50 | 33.72 | 34.50 | 55.07 | 31.69 | 20.66 | 47.06 | 56.30 | 46.58 |
| exp5 | 48.96 | 1,363.26 | 60.48 | 63.18 | 55.92 | 70.35 | 35.70 | 31.40 | 57.19 | 32.88 | 34.60 | 54.74 | 32.10 | 22.70 | 47.30 | 58.83 | 57.75 |

Table 18: **All Benchmark Results for Data Ratio Experiments with fixed 1350k data**

### G.5  737K and 5M Mixes

**0.7M** For the 0.7M data we used in Section 2.4, We add a small number of OCR and chart data to LLaVA 665K, specifically 15,501 AI2D, 14,999 DocVQA, and 13,000 DVQA data points. This results in a 737K mix, which covers all categories in training MLLMs. This data mix allows us to study visual representations efficiently.

**5.0M** For the 5M data mixes we use in Section 2.4, we apply data filtering discussed in Section 4.2 and apply $t$=150k on all multimodal instruction data in Cambrian-10M.

### G.6  Test Image Leakage in Visual Instruction Training Data

One potential concern with our targeted data engine (Section 4.1) is that instruction-tuning data collected from the open web could introduce data leakage. To address this, we systematically analyze the extent of direct image matches between our training data and our test sets. Using difference hashing (dHash) [19], we compute hashes for all images in the training data and test sets. We then compare these hash sets to determine how many test images overlap with our training data, reporting the number of collisions in Table 19.

Across all fifteen datasets, our targeted data engine finds only 32 test images in total, amounting to just 0.06% of the test data. This low overlap percentage dispels concerns that our data engine inadvertently targets specific test sets. When analyzing the full Cambrian10M dataset—which is 15x larger than LLaVA-665k—we observe only 6x more matching test images (7,244 compared to 1,034 in LLaVA-665k). This discrepancy suggests that Cambrian10M's scale does not inherently result in excessive overlap with test sets. Instead, any overlap likely arises from the natural reuse of training images across benchmark datasets rather than targeted duplication.

| Category | Test Set | # Images | Data Eng. | Cambrian10M | LLaVA-665k |
|---|---|---|---|---|---|
| General | MME$^P$ | 2,374 | 0 (0.00%) | 332 (13.98%) | 82 (3.45%) |
| | MMB | 4,377 | 7 (0.16%) | 1,122 (25.63%) | 533 (12.18%) |
| | SEED$^I$ | 17,990 | 6 (0.03%) | 26 (0.14%) | 0 (0.00%) |
| | GQA | 398 | 0 (0.00%) | 1 (0.25%) | 0 (0.00%) |
| Knowledge | SQA$^I$ | 2,017 | 0 (0.00%) | 1,263 (62.62%) | 0 (0.00%) |
| | MMMU$^V$ | 900 | 1 (0.11%) | 3 (0.33%) | 0 (0.00%) |
| | MathVista$^M$ | 1,000 | 2 (0.20%) | 259 (25.90%) | 15 (1.50%) |
| | AI2D | 3,088 | 0 (0.00%) | 1,458 (47.22%) | 0 (0.00%) |
| OCR & Chart | ChartQA | 2,500 | 14 (0.56%) | 670 (26.80%) | 0 (0.00%) |
| | OCRBench | 1,000 | 0 (0.00%) | 177 (17.70%) | 59 (5.90%) |
| | TextVQA | 5,000 | 2 (0.04%) | 1,122 (22.44%) | 9 (0.18%) |
| | DocVQA | 5,188 | 0 (0.00%) | 53 (1.02%) | 0 (0.00%) |
| Vision-Centric | MMVP | 300 | 0 (0.00%) | 0 (0.00%) | 0 (0.00%) |
| | RealWorldQA | 765 | 0 (0.00%) | 0 (0.00%) | 0 (0.00%) |
| | CV-Bench | 2,638 | 0 (0.00%) | 758 (28.73%) | 336 (12.74%) |
| **Total** | | 49,535 | 32 (0.06%) | 7,244 (14.62%) | 1,034 (2.07%) |

Table 19: **Number of leaked test set *images***. Using image hashing, we assess the overlap of test images across three training datasets: *Cambrian10M Data Engine 161k subset ("Data Eng.")*, *Cambrian10M*, and *LLaVA-665k*. We list the number of images in each test set, as well as the number of matching images and percentage of overlap for each training set in blue. Our Data Engine finds a neglible 0.06% of test images, dispelling any concerns that it is targeting the test sets. The full Cambrian10M training set contains 7,244 test set images, whereas LLaVA-665k contains 1,034. Despite being a 15x larger dataset, Cambrian10M only has 6x more overlapping images. Such overlap is inevitable since many test sets use validation images from standard benchmarks (like COCO). It is worth highlighting: **although exact image matches are found, this does not mean that exact image-question pairs have been found.** Unlike in prior *unimodal* paradigms of computer vision research, in the *multimodal* setting, a single data point is composed of an image-text (question) pair, not just the image itself. Thus, seeing a test image during training is not equivalent to "training on the test set" so long as the training image does not have the same text pair as the test data point.

It is important to emphasize that while some exact image matches are found, this does not imply that the exact image-question pairs have been encountered during training. Unlike in traditional unimodal computer vision research, where an image alone constitutes a data point, the multimodal paradigm treats each image-text (question-answer) pair as unique. Consequently, seeing a test image during training is not equivalent to "training on the test set" as long as the associated text (question-answer) pairs differ. This distinction ensures that Cambrian10M respects the integrity of test evaluations, even in cases where images might appear in both training and test sets.

We encourage future research exploring the impact of image-only leakage on the performance of MLLMs. Understanding this influence may yield insights into the boundaries of model generalization and guide future best practices for dataset construction in multimodal learning.

### G.7 Broader Impacts

We conducted a preliminary analysis of the Cambrian dataset, focusing on the distribution of male, female, and neutral pronouns. Our findings show the following distribution: 38.35% male pronouns, 17.99% female pronouns, and 43.66% neutral pronouns.

We recognize that training models on biased data can perpetuate these biases. Addressing bias by artificially modifying data distributions—such as through rebalancing or applying fairness constraints—can help mitigate this issue, but it also presents challenges. These include the potential loss of generalization and the risk of introducing new biases. Additionally, identifying and mitigating bias in Multimodal Large Language Models (MLLMs) is particularly complex, given the interaction

between different data modalities. We believe that openness in model development and data curation will accelerate research aimed at understanding and mitigating these potential harms.

## H   Implementation Details

**Cambrian Models** For our final Cambrian models, we use 2.5M adapter data which is comprised of 1.2M captioning data from shareGPT4V [27] and 1.2M captioning data used in MiniGemini [77].

**SVA** We provide here ablation studies of SVA module.

| $D$ | OCR & Chart | | $G$ | OCR & Chart | | Multi-agg | OCR & Chart |
|---|---|---|---|---|---|---|---|
| 2 | 52.1 | | 1 | 52.4 | | No | 52.4 |
| 3 | 52.4 | | 2 | 52.6 | | Yes | **53.3** |
| 4 | **52.8** | | 3 | **53.1** | | | |
| | (a) # layers | | | (b) # groups | | | (c) Multi-layer aggregation |

Table 20: **Ablations on hyperparameter choices for SVA.** Enlarging the model capacity of the SVA module can further improve the performance.

We further conduct ablation experiments using OpenAI CLIP ViT-L/14@336 + OpenCLIP ConvNeXt-L@1024 as our base model combination. We focus on the OCR & chart categories to assess the impact on high-resolution visual understanding. The results show that increasing capacity via $D$ or $G$ improves performance and that allowing vision aggregation across multiple layers by adding cross-attention layers within the LLM also enhances performance.

Compared with other spatial-based connectors like C/D-Abstractor [21] which are designed for single vision feature maps, our SVA module can dynamically combine visual features from multiple vision models with varying resolutions. Besides, our spatial inductive bias in SVA can better compress spatial information compared with such methods. To isolate the effect of spatial inductive bias, we consider the case of token reduction using a single vision encoder. Specifically, we use OpenAI CLIP ViT-L as the vision model and compress its original 576 tokens to 36 tokens using our SVA module and other connectors. We compare our SVA module with three baselines: 1) Direct interpolation + MLP, 2) C-Abstractor [21], and 3)LDPv2 Projector [35] (similar to C-Abstractor but more lightweight). For fair comparisons, we do not include multi-layer aggregation inside the LLM for our SVA baseline, and the results are shown in Table 21. Compared with the simple MLP baseline, C-Abstractor performs better on General and Vision-Centric tasks but inferior on Knowledge and OCR & Chart tasks. LDPv2 performs similarly to the MLP baseline. Our SVA consistently demonstrates superior performance across all categories, especially in OCR & Chart and Vision-Centric tasks, demonstrating its effectiveness in information compression.

| Method | General | Knowledge | OCR & Chart | Vision-Centric |
|---|---|---|---|---|
| Interpolate + MLP | 63.4 | 43.8 | 28.1 | 43.7 |
| C-Abstractor [21] | 64.4 | 42.8 | 26.1 | 44.3 |
| LDPv2 [35] | 62.5 | 43.9 | 28.7 | 43.9 |
| **SVA** | **65.5** | **44.5** | **31.4** | **46.9** |

Table 21: **Comparison between SVA and other spatial-based connectors vision token compression.** The SVA module with spatial inductive bias more effectively compresses the vision information.

We introduce learnable $k_m \times k_m$ positional encodings in the vision features when $k_m > 1$. Besides, during cross-attention, the query is augmented with a global feature obtained by global pooling over the vision features, which is concatenated with $\mathbf{q}_{i,j}$ to better guide the aggregation process. In our experiments, the feature maps of all vision encoders except for ConvNext are interpolated to 576×576 ($m_k = 1$ for $L = 24$). For ConvNext, we first interpolate the feature maps from its 4 stages to $96 \times 96$ ($m_k = 4$ for $L = 24$) and then channel-wise concatenate them to form its final vision feature map similar to [77].

For experiments in Section 3, we set $D = 3$, $G = 1$ and add cross-attention layers between the layers of LLM with a stride equal to 3. For our final Cambrian models, we set $D = 3$, $G = 1$ and insert multiple cross-attention layers in LLM considering the tradeoff between performance and efficiency. For Cambrian-8B, Cambrian-13B, and Cambrian-34B, the strides of cross-attention layers inside the LLM are 3, 4, and 9 respectively.

To study the importance of visual features from different vision models to different image categories, we further investigate the attention score distribution in our SVA module. We evaluate our Cambrian-8b model on GQA, DocVQA, and ScienceQA (representing three different benchmark categories), and the attention distribution results are shown in Table 22. We can see that on real-world images (GQA), the contribution of different vision models is relatively uniform, in part due to the similar characteristics of SigLIP and CLIP. On document-type images (DocVQA) which are text-heavy and often high-resolution, the influence of SigLIP increases and that of ConvNext greatly increases to aid in high-resolution information processing. For scientific images (ScienceQA) composed of illustrations and diagrams about different science categories, the contribution of SigLIP is further increased while the portion of DINOv2 decreases compared to GQA.

| Model | GQA | DocVQA | ScienceQA |
|---|---|---|---|
| SigLIP | 29.7% | 31.1% | 35.2% |
| CLIP | 18.5% | 13.4% | 16.3% |
| DINOV2 | 24.1% | 11.0% | 17.6% |
| ConvNext | 27.7% | 44.5% | 30.9% |

Table 22: **Attention distribution studies.** The attention distribution among different vision encoders varies with different image categories.

**Unfreezing**  While unfreezing is largely beneficial (Section 2.3 and Fig. 13), it has a significant speed drawback. Given fixed computational resources, unfreezing visual encoders slows down the fine-tuning process by approximately 50–55%. For initial explorations or when computational overhead is a concern, leaving the visual encoders frozen can be a practical strategy. This allows for quicker iterations and tuning, especially during early research phases, while still providing valuable insights. Ultimately, unfreezing is recommended for achieving the best performance once the setup has been optimized.

# I  Evaluation Details

## I.1  System Prompts Used in Evaluation

To ensure the reproduction of our results, we also include the system prompts we used in this work. The system prompts for our models can be found in Table 25. Additionally, we release the prompts we used while evaluating our models on the various benchmarks in Table 26. We hope this sets a precedent for future research to improve the reproducibility of benchmark results.

## I.2  Ablation Study on Fuzzy Matching Vs LLM Judgement

We use fuzzy matching to evaluate responses in some benchmarks, since MLLMs can answer questions with auxillary phrases. To study the effectiveness of our fuzzy matching, we compare our model accuracy through fuzzy matching with the model accuracy obtained when we use LLM as a grader.

The LLM grader is sensitive to the prompt given to it while grading, and we prompt the LLM (we use OpenAI GPT-3.5-turbo and GPT-4-turbo as our graders) with few shot grading examples, which we notice significantly improves grading accuracy. An example of such a prompt is given below.

> **LLM Grader Prompt**
>
> ```
> You are a reliable grader. Reply with only either of the following
> 2 words: CORRECT or INCORRECT.
> ```

| Experiment | LLM | Backbone Vision | Data Adapter | Data Instruction Tuning | Adapter lr | Adapter wd | Adapter bs | Instruction Tuning lr | Instruction Tuning wd | Instruction Tuning bs | vision lr |
|---|---|---|---|---|---|---|---|---|---|---|---|
| 0M Adapter+737K IT | Vicuna-1.5-7B | OpenAI CLIP ViT-L/14@336 | 0 | 737k | - | - | - | 2e-5 | 0 | 512 | - |
| 0.5M Adapter+737K IT | Vicuna-1.5-7B | OpenAI CLIP ViT-L/14@336 | 0.5M | 737k | 1e-3 | 0 | 512 | 2e-5 | 0 | 512 | - |
| 1.2M Adapter+737K IT | Vicuna-1.5-7B | OpenAI CLIP ViT-L/14@336 | 1.2M | 737k | 1e-3 | 0 | 512 | 2e-5 | 0 | 512 | - |
| Unfreeze Vision | Vicuna-1.5-7B | OpenAI CLIP ViT-L/14@336 | 1.2M | 737k | 1e-3 | 0 | 512 | 2e-5 | 0 | 512 | 1e-5 |
| Model Ensemble | Vicuna-1.5-7B | Chosen Combination | 1.2M | 737k | 1e-3 | 0 | 512 | 2e-5 | 0 | 512 | |
| Data Balance | Vicuna-1.5-7B | OpenAI CLIP ViT-L/14@336 | 0 | Mix Based on three $t$ | - | - | - | 2e-5 | 0 | 512 | - |
| Data Ratio | Vicuna-1.5-7B | OpenAI CLIP ViT-L/14@336 | 0 | 1350k Based on Ratio | - | - | - | 2e-5 | 0 | 512 | - |
| LLaVA 665K | Vicuna-1.5-7B | OpenAI CLIP ViT-L/14@336 | 0 | LLaVA 665K | - | - | - | 2e-5 | 0 | 512 | - |
| Cambrian-10M (Data) | Vicuna-1.5-7B | OpenAI CLIP ViT-L/14@336 | 0 | Cambrian-10M | - | - | - | 2e-5 | 0 | 512 | - |
| Cambrian-7M (Data) | Vicuna-1.5-7B | OpenAI CLIP ViT-L/14@336 | 0 | Cambrian-7M | - | - | - | 2e-5 | 0 | 512 | - |
| Cambrian-1-8B | Llama-3-Ins-8B | SVA with 4 encoders* | 2.5M | Cambrian-7M | 1e-4 | 0 | 512 | 2e-5 | 0 | 512 | - |
| Cambrian-1-13B | Vicuna-1.5-13B | SVA with 4 encoders* | 2.5M | Cambrian-7M | 1e-4 | 0 | 512 | 2e-5 | 0 | 512 | - |
| Cambrian-1-34B | Hermes-2-Yi-34B | SVA with 4 encoders* | 2.5M | Cambrian-7M | 1e-4 | 0 | 512 | 2e-5 | 0 | 1024 | - |

Table 23: **Implementation details and hyperparameters for all experiments.** *4 encoders are: OpenAI CLIP ViT-L/14@336, SigLIP ViT-SO400M/14@384, DINOv2 ViT-L/14@518, Open-CLIP ConvNeXt-XXL@1024

```
You will be given an 'answer' and a 'gt_answer' (ground truth answer)
,and you must reply with either CORRECT or INCORRECT based on the
response. Tolerate a 0.05 relative error for numerical answers.
answer: 25
gt_answer: 29
evaluation: INCORRECT
answer: Yes
gt_answer: Yes
evaluation: CORRECT
answer: 80
gt_answer: 80
evaluation: CORRECT
answer: Ireland
gt_answer: Italy
evaluation: INCORRECT
answer: UK
gt_answer: UK
evaluation: CORRECT
answer: 2019
gt_answer: 2011
evaluation: INCORRECT
answer: {answer}
gt_answer: {gt_answer}
evaluation:
```

We conduct an ablation study on the benchmarks that require fuzzy matching and present the results in Table 24. We discover that fuzzy matching provides reliable results compared to an LLM grader. We recommend using a more capable model (such as GPT-4-turbo) for grading benchmarks that have more subjective responses (such as numbers and words).

| Method | Eval | SEED[I] | GQA | SQA[I] | MMMU | AI2D | ChartQA | OCRBench | TextVQA | MMVP | RealWorldQA |
|---|---|---|---|---|---|---|---|---|---|---|---|
| Cambrian-1 8B | Fuzzy Matching | 74.7 | 64.6 | 80.4 | 42.7 | 73.0 | 73.3 | 62.4 | 71.7 | 51.3 | 64.2 |
| Cambrian-1 8B | GPT3.5 Grading | 78.4 | 65.8 | 82.0 | 38.9 | 78.1 | 71.2 | 67.0 | 69.2 | 49.3 | 63.5 |
| Δ | | +3.7 | +1.2 | +1.6 | -3.8 | +5.1 | -2.1 | +4.6 | -2.5 | -2.0 | -0.7 |
| Cambrian-1 13B | Fuzzy Matching | 74.7 | 64.6 | 80.4 | 40.4 | 73.0 | 73.3 | 62.4 | 71.7 | 51.3 | 64.2 |
| Cambrian-1 13B | GPT3.5 Grading | 77.3 | 64.7 | 81.3 | 37.2 | 78.2 | 71.4 | 67.1 | 75.6 | 46.0 | 64.3 |
| Δ | | +2.9 | +0.4 | +2.0 | -3.2 | +4.6 | -2.4 | +5.2 | +2.8 | +4.7 | +1.3 |

Table 24: **Comparison between Fuzzy Matching Accuracy and LLM Judged Accuracy.** Fuzzy matching and LLM referee yield similar accuracies for the benchmarks that require matching.

| LLM Backbone | System Prompt |
|---|---|
| Vicuna 1.5 7B | A chat between a curious user and an artificial intelligence assistant. The assistant gives helpful, detailed, and polite answers to the user's questions. |
| LLAMA-3 8B | You are Cambrian, a highly intelligent multimodal AI trained by NYU Vision X. As a multimodal AI, you have the ability to process and analyze images. Whenever an image is present in the conversation, very carefully examine it and consider its content when formulating your response.You should give concise responses to very simple questions, but provide thorough responses to more complex and open-ended questions. |
| Nous-Yi 34B | You are Cambrian, a highly intelligent multimodal AI trained by NYU Vision X. As a multimodal AI, you have the ability to process and analyze images. Whenever an image is present in the conversation, very carefully examine it and consider its content when formulating your response. You should give concise responses to very simple questions, but provide thorough responses to more complex and open-ended questions. |

Table 25: LLM Backbone System Prompts

Table 26: Listing the prompts used in the evaluation of each benchmark

| Benchmark | Prompt | Example |
|---|---|---|
| AI2D | \nAnswer with the option's letter from the given choices directly. | USER: <image>\nwhich of these define dairy item\n(A) c\n(B) D\n(C) b\n(D) a\nAnswer with the option's letter from the given choices directly. ASSISTANT: |
| ChartQA | \nAnswer the question using a single number or phrase. | USER: <image>\nHow many food item is shown in the bar graph?\nAnswer the question using a single number or phrase. ASSISTANT: |
| DocVQA | \nGive the short answer directly. | USER: <image>\nWhat is the dividend payout in 2012?\nGive the short answer directly. ASSISTANT: |
| GQA | \nAnswer the question using single word or phrase. | USER: <image>\nIs it overcast?\nAnswer the question using single word or phrase. ASSISTANT: |
| MathVista | \nFirst show your reasoning process and then give the final answer. | USER: <image>\nwhat is the total volume of the measuring cup?\nFirst show your reasoning process and then give the final answer. ASSISTANT: |
| MM-Bench EN | \nAnswer with the option's letter from the given choices directly. | USER: <image>\nWhich of the following was a dependent variable in this experiment?\n(A) cocoon\n(B) chrysalis\n(C) nan\n(D) nan\nAnswer with the option's letter from the given choices directly. ASSISTANT: |
| MME | \nPlease answer the question using a single word or phrase. | USER: <image>\nIs a python code shown in the picture? Please answer yes or no.\nAnswer the question using a single word or phrase. ASSISTANT: |
| MMMU | \nAnswer with the option's letter from the given choices directly. | USER: <image>\nWhat causes these unusual formations on Mountain papaya? Options:\nA. Abiotic\nB. Confused\nC. Biotic\nD. Normal\nAnswer with the option's letter from the given choices directly. ASSISTANT: |
| MMVP | \nAnswer with the option's letter from the given choices directly. | USER: <image>\nAre the butterfly's wings closer to being open or closed? Options:\n(a) Open\n(b) Closed\nAnswer with the option's letter from the given choices directly. ASSISTANT: |
| OCR Bench | \nGive the short answer directly. | USER: <image>\nwhat is written in the image?\nGive the short answer directly. ASSISTANT: |
| RealWorld QA | \nAnswer the question using a single word or phrase. | USER: <image>\nIn which direction is the front wheel of the car on the right side facing?\n\nA. Left\nB. Straight\nC. Right\nAnswer the question using a single word or phrase. ASSISTANT: |
| SQA-I | \nAnswer with the option's letter from the given choices directly. | USER: <image>\nWhat is the name of the colony shown?\nA. Maryland\nB. New Hampshire\nC. Rhode Island\nD. Vermont\nAnswer with the option's letter from the given choices directly. ASSISTANT: |
| SEED-I | \nAnswer with the option's letter from the given choices directly. | USER: <image>\nHow many towels are in the image? Options:\nA. One\nB. Two\nC. Three\nD. Four\nAnswer with the option's letter from the given choices directly. ASSISTANT: |
| Text-VQA | \nAnswer the question using a single word or phrase. | USER: <image>\nwhat is the time?\nReference OCR tokens: N, u, g0\nAnswer the question using a single word or phrase. ASSISTANT: |
| ADE | \nAnswer with the option's letter from the given choices directly. | USER: <image>\nConsidering the relative positions of the cushion and the sofa in the image provided, where is the cushion located with respect to the sofa? Select from the following choices.\n(A) right\n(B) left\nAnswer with the option's letter from the given choices directly. ASSISTANT: |

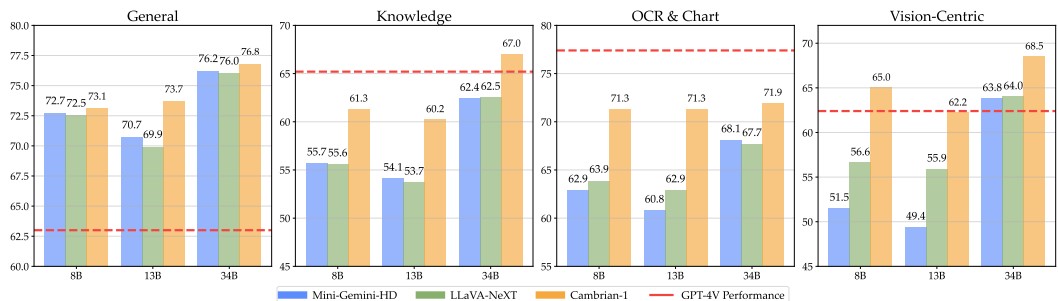

Figure 18: **Comparison of model average performances on each category.** Cambrian-1 outperforms other open-source models across all sizes. The lead is especially large on OCR & Chart and Vision-Centric benchmarks, highlighting the advantage of our vision-centric design.

| Benchmark | Prompt | Example |
|---|---|---|
| COCO | \nAnswer with the option's letter from the given choices directly. | USER: <image>\nHow many trains are in the image? Select from the following choices. \n(A) 3\n(B) 0 \n(C) 1 \n(D) 2 \n(E) 4\nAnswer with the option's letter from the given choices directly. ASSISTANT: |
| Omni3D | \nAnswer with the option's letter from the given choices directly. | USER: <image>\nEstimate the real-world distances between objects in this image. Which object is closer to the traffic cone (highlighted by a red box), the motorcycle (highlighted by a blue box) or the bus (highlighted by a green box)?\n(A) motorcycle\n(B) bus\nAnswer with the option's letter from the given choices directly. ASSISTANT: |

## J   More results of Cambrian-1 Model

In Fig. 18, we also plot our Cambrian-1 performance as well as GPT-4 performances. In the plot, it is clear that Cambrian-1 offers competitive performance compared to proprietary models in most categories. We also showcase some examples in Fig. 19, demonstrating that the model effectively attends to details in images despite using only 576 tokens.

## K   Potential Misuse & Mitigation Strategies

We recognize that there are ethical concerns regarding the potential misuse of multimodal large language models like Cambrian-1, particularly in generating misleading content or spreading misinformation. Below, we outline the main risks and provide strategies to address them:

1. **Misinformation** Cambrian-1 could be used to create misleading text descriptions of images, leading to false narratives or misrepresentations. For instance, such models might be leveraged by social media bots to manipulate public opinion during elections or other critical events.

2. **Hallucination** Similar to any large language model, Cambrian-1 may produce information that is not based on facts or actual input data. This phenomenon, often called "hallucination," can be dangerous if users assume the model's output is entirely accurate without verification.

To mitigate these risks, users should exercise caution and critical thinking when interpreting outputs generated by Cambrian-1. It is important to verify the information produced by the model, particularly if the results are intended for sensitive or high-stakes applications. Users must be aware of the potential for hallucinations, where the model produces information not grounded in facts, and take steps to cross-check and validate any critical outputs. Additionally, implementing content filtering as a safeguard can help flag potentially harmful or misleading content before it is disseminated.

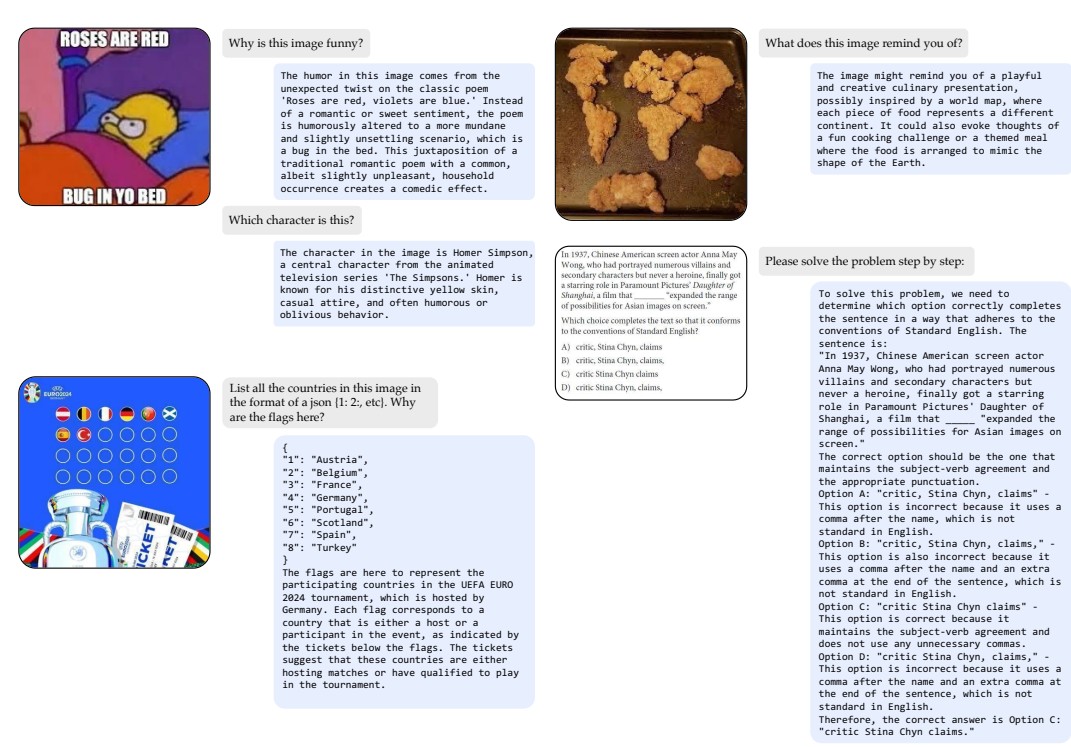

Figure 19: **Examples of Cambrian-1-34B.** Cambrian-1 showcases impressive abilities in visual intersection. The model demonstrates instruction-following ability such as output in json format, as illustrated in the bottom-left example. Cambrian-1 also demonstrates remarkable OCR ability (See model handles different Comma "," in the right down example).

