# OpenReview forum: "Cambrian-1: A Fully Open, Vision-Centric Exploration of Multimodal LLMs"
_NeurIPS.cc/2024/Conference — NeurIPS 2024 oral_

### Official Review · Reviewer_ED7N · 2024-06-17

**Soundness:** 3
**Presentation:** 3
**Contribution:** 3
**Rating:** 6
**Confidence:** 5

**Summary:**

This paper presents extensive studies on existing MLLM benchmarks addressing the difficulties involved in consolidating and interpreting results from various tasks for MLLM designs. Moreover, the authors also propose Spatial Vision Aggregator (SVA), a dynamic and spatially-aware connector to fuse vision features with LLMs while reducing vision tokens. In addition, authors also collect high quality visual instruction-tuning data. The proposed model, Cambrian-1, achieves the state-of-the-art performance on multiple MLLM benchmarks.

**Strengths:**

1, Overall writing is good and easy to follow. The motivation is clear.

2, The goal of this work is interesting. A good study for the effects on Large Language Model, Visual Encoder, Multimodal Connector, Data Curation Pipeline, Instruction in the MLLM system.

3, Experiments are extensive, and several findings are useful when designing MLLM models.

4, A new dataset Cambrian-7M is proposed, which may benefit the MLLM fields for further research.

5, The performance looks good compared with LLaVA-Next. If open-sourced, this model can definitely benefit for the community.

6, Open-sourced code and model.

**Weaknesses:**

1, The technical novelty is limited, combining multiple vision experts into MLLM is not new. Moreover, fusing visual tokens dynamically is also not new in dynamic network design.

[-] Sphinx: the joint mixing of weights, tasks, and visual embeddings for multi-modal large language models, arixv-2023.

[-] Mova: Adapting mixture of vision experts to multimodal context, arxiv-2024.

I think there are more close related works than [1][2]. The authors should cite all these works as respect.


2, I also have one question, What if you do not use the dynamic token but only using proposed tuning datasets, compared with LLaVA-1.6.

3, Several findings are well known and also verified from previous works. For example, Finding-6 is common knowledge using high resolution visual encoder.

[-] Vary: scaling up the vision vocabulary for large vision-language models


4, Several figures are useless and common knowledge for MLLM community. For example, Fig.1 and Fig.2 can be merged into one figure.

5, The ablation studies for SVA are not enough. For example, which tokens are more important in which datasets? This needs further analysis. Moreover, the effect of increasing instruction tuning data size is not well explored.

Given these, I rate this work as weak accept.

**Questions:**

See the weakness part.

---

> ### Author Rebuttal · Authors · 2024-08-06
>
> We thank the reviewer for the thorough review and acknowledgments. We appreciate that you find our work is “well-written”, contains “extensive experiments” and “good performance”, and will “definitely benefit the community” when open-sourced. We summarize your questions and provide responses below:
>
> > **W1&3: Findings are well-known and verified from previous work**
>
> A: We believe the study of the Multimodal Large Language Model contains many moving parts, and previous works provide isolated studies on each component. This potentially leads to contradictory results in different studies (e.g., freeze or unfreeze vision encoder in Prismatic VLM vs. Deepseek VL). Our work undertakes a *systematic study*, isolating and studying the numerous moving parts, which we believe allows more long-standing conclusions that benefit the community.
>
> Further, we respectfully argue that our work differs from previous works in the following ways:
>
> - *More Comprehensive Experiments*: Our study considers more vision backbones and organized benchmarks than previous works, yielding new insights. For example, we find that high-resolution ConvNets benefit OCR & Chart tasks, and self-supervised learning models can potentially compete with language-supervised ones. Our experiments also reveal the properties of different CLIP models (e.g., SigLIP, EVA-CLIP) beyond ImageNet accuracy, providing valuable insights for both MLLM and Vision model developers. For instance, we observe that EVA-CLIP performs well in most domains but struggles with Chart tasks. This highlights the need for CLIP developers to focus more on OCR & Chart data collection during training.
> - *Findings on high-res encoders*: While concurrent works emphasize the importance of high-resolution features, we take a step further to pinpoint the potential of using ConvNets, such as the ConvNext OpenCLIP model, to efficiently and effectively process high-resolution images.
> - *New Vision Model Aggregation Strategy*: Compared to previous works that focus on fusing vision models, we study both *which models to combine* (§3.5) and *strategies for combining them* (§4). Our SVA approach preserves the resolution, maintains the spatial inductive bias in images, and uses a fixed number of visual tokens.
>
> We also thank the reviewer for raising this point and suggesting these works. We will discuss each of them in our revision.
>
> > **W2: Training only using proposed tuning datasets, compare with LLaVA 1.6**
>
> A: We first want to clarify that LLaVA 1.6 (LLaVA-Next) proposes a dynamic resolution approach using *2880* tokens, while our SVA module uses only *576* fixed tokens. We conduct additional experiments with the LLaVA model trained using LLaMA-3-8b and Cambrian-7M data. Due to the short rebuttal period, we use the conventional 576 visual tokens in LLaVA and LLaVA-1.5, not the dynamic high-resolution proposal of LLaVA-Next. Despite fewer tokens, this version matches or outperforms LLaVA-Next in General, Knowledge, and Vision-Centric tasks. Adding our SVA module further improves performance, especially in OCR & Chart tasks, while still using only 576 tokens.
>
> |Data|# of Vis Tokens|General|Knowledge|OCR & Chart|Vision-Centric|
> |:--|--:|--:|--:|--:|--:|
> |LLaVA-Next|2880|72.5|55.6|61.0|55.0|
> |LLaVA w/ Cambrian Data|576|72.0|58.1|54.3|55.6|
> |Cambrian-8B|576|74.4|60.1|66.2|60.3|
>
> > **W4: Several figures are useless and common knowledge for MLLM (e.g, Figs. 1 & 2)**
>
> A: We hope our work provides a systematic study around MLLMs and can serve as informational for audiences both within and beyond the MLLM community. Especially now, as MLLM is becoming an ever-growing community, the introduction and figures serve as preparation and context-setting for a broader audience. We make no claims of novel findings in the initial figures and reserve such insights for the later sections after providing the audience requisite context. Nevertheless, we thank the reviewer for raising this concern, and we will consider condensing our presentation in the revised version.
>
> > **W5: Study of SVA Module is not enough**
>
> A: We thank the reviewer for raising this crucial point. We have added a study of the importance of visual features from different vision models to different image categories by investigating the attention score distribution in our SVA module.
>
> We evaluate our Cambrian-8b model on GQA, DocVQA, and ScienceQA (representing three different benchmark categories), and tabulate attention distributions below. We can see that on real-world images (GQA), the contribution of different vision models is relatively uniform, in part due to the similar characteristics of SigLIP and CLIP. On document-type images (DocVQA) which are text-heavy and often high-resolution, the influence of SigLIP increases and that of ConvNext greatly increases to aid in high-resolution information processing. For scientific images (ScienceQA) composed of illustrations and diagrams about different science categories, the contribution of SigLIP is further increased while the portion of DINOv2 decreases compared to GQA.
>
> |Model|GQA|DocVQA|ScienceQA|
> |:--|--:|--:|--:|
> |SigLIP|29.7%|31.1%|35.2%|
> |CLIP|18.5%|13.4%|16.3%|
> |DINOv2|24.1%|11.0%|17.6%|
> |ConvNext|27.7%|44.5%|30.9%|
>
> We also study the performance of our Cambrian-8b model with SVA modules on different sizes of alignment and instruction tuning data. The results are shown below. We can see that increasing the size of alignment data leads to improvement in all benchmark categories. Increasing the size of instruction tuning data leads to notable overall improvement, and the instruction tuning data is especially helpful for Knowledge, OCR & Chart, and Vision-Centric tasks.
>
> ||General|Knowledge|OCR & Chart|Vision-Centric|
> |:--|--:|--:|--:|--:|
> |1.2M alignment + 0.7M instruction|72.3|54.8|58.3|57.2|
> |2.5M alignment + 0.7M instruction|72.7|55.8|58.9|58.3|
> |2.5M alignment + 7M instruction|74.4|60.1|66.2|60.3|

---

> > ### Comment · Reviewer_ED7N · 2024-08-12
> > **Rebuttal comments**
> >
> > Thanks for your reply. The rebuttal solves my concern. I keep original rating as weak accept.

---

### Official Review · Reviewer_K2Un · 2024-07-13

**Soundness:** 3
**Presentation:** 4
**Contribution:** 3
**Rating:** 7
**Confidence:** 5

**Summary:**

This paper introduces a multimodal large language models (MLLMs), named Cambrian-1, designed with a vision-centric approach. In current MLLM researches, the choices of visual encoder are not sufficiently explored. This study utilizes MLLM performance as a visual representation evaluator, showing different characteristics over differently trained vision encoders and revealing that various widely-used MLLM benchmarks are disconnected from visual understanding capability but connected to language capability. Furthermore, this study proposes spatial vision aggregator (SVA) to effectively connect vision and language models with spatial inductive bias. Additionally, curation of high-quality visual instruction-tuning dataset and its distribution balancing are discussed. As a result, Cambrian-1 achieves state-of-the-art performances and provides an open cookbook for MLLMs.

**Strengths:**

- This paper is notably well-written and easy to follow.
- Section 3.1 shows the limitations of MLLM benchmarks. The finding that several existing benchmarks like MMMU, which were considered important benchmarks in the MLLM field, do not properly evaluate multimodal capabilities is very interesting.
- This study releases model weights, code, tools, datasets, and detailed recipes, which is a great contribution to this field.

**Weaknesses:**

- There exists a previous work about vision-language connectors with spatial inductive bias [1]. The comparison or at least discussion between the proposed SVA and C/D-Abstractor [1] is essential but lacks.
- There are many overlapping findings with existing studies. For example, language-supervised models are effective [2], high-res encoders are beneficial [3], increasing data size and spatial inductive bias are advantageous for connectors [1], and so on. I believe that re-examining these aspects and analyzing them in different settings has its own contribution due to the empirical nature of this field. Nevertheless, it is difficult to attribute high value to the overlapping findings.
- Findings 7 (the second Findings 6 in the paper, seems to be a typo) is not consistent with the results. The finding claims that performance improves with the vision encoder ensemble, but Table 11 does not seem to support this. For example, SigLIP+DINOv2 performs worse than sole SigLIP.

[1] Cha, Junbum, et al. "Honeybee: Locality-enhanced projector for multimodal llm." Proceedings of the IEEE/CVF Conference on Computer Vision and Pattern Recognition. 2024.
[2] Chen, Xi, et al. "Pali-3 vision language models: Smaller, faster, stronger." arXiv preprint arXiv:2310.09199 (2023).
[3] Liu, Haotian, et al. "Improved baselines with visual instruction tuning." Proceedings of the IEEE/CVF Conference on Computer Vision and Pattern Recognition. 2024.

**Questions:**

In Table 11, why are there two entries for "SigLIP+DINOv2+ConvNext" with different numbers?

**Limitations:**

This study construct a new dataset based on web search, but it does not appear to address any privacy issues. It would be better to address this issue.

---

> ### Author Rebuttal · Authors · 2024-08-06
>
> We thank the reviewer for the thorough review and acknowledgments. We appreciate that you find our work “notably well-written”, “shows the limitations of MLLM benchmarks”, and “a great contribution to this field” via our fully-open approach. We summarize your questions and provide our explanations below:
>
> > **W1: Comparison and Discussion between SVA and C/D-Abstractor**
>
> A: We appreciate the valuable suggestion. We provide such discussion and analyses here, and will incorporate this into our revision.
>
> We first emphasize that our SVA module is distinct from other spatial-based connectors (e.g., C/D-Abstractor) in its ability to dynamically combine visual features from *multiple* vision models with *varying* resolutions.
>
> However, to isolate the effect of spatial inductive bias, we consider the case of token reduction using a single vision encoder. Specifically, we use OpenAI CLIP as the vision model and compress its original 576 tokens to 36 tokens using our SVA module and other connectors. We include three baselines:
>
> 1. Direct interpolation + MLP
> 2. C-Abstractor
> 3. LDPNetV2Projector (similar to C-Abstractor but more light-weight)
>
> We conduct experiments with the 1.2M alignment + 0.7M instruction tuning data setting with Vicuna-1.5-7b as the language model. For fair comparison, we do not include multi-layer aggregation inside the LLM for our SVA baseline.
>
> We tabulate results below:
>
> | Method            | General | Knowledge | OCR & Chart | Vision-Centric |
> |-------------------|:-------:|:---------:|:-----------:|:--------------:|
> | Interpolate + MLP |  63.4   |   43.8    |    28.1     |      43.7      |
> | C-Abstractor      |  64.4   |   42.8    |    26.1     |      44.3      |
> | LDPNetV2          |  62.5   |   43.9    |    28.7     |      43.9      |
> | SVA               |**65.5** | **44.5**  |  **31.4**   |    **46.9**    |
>
> Compared with the simple MLP baseline, C-Abstractor performs better on General and Vision-Centric tasks but inferior on Knowledge and OCR & Chart tasks. LDPNetV2 performs similarly to the MLP baseline. Our SVA consistently demonstrates superior performance across all categories, especially in OCR & Chart and Vision-Centric tasks, demonstrating its effectiveness in information compression.
>
> One possible explanation for SVA's data efficiency compared to C-Abstractor is that the SVA module performs local attention on all positions in the grid with the same parameters, so our SVA module receives more supervision.
>
>
> > **W2: Overlapping findings with existing studies**
>
> A: We believe the study of the Multimodal Large Language Model contains many moving parts, and previous works provide isolated studies on each component. This potentially leads to contradictory results in different studies (e.g., freeze or unfreeze vision in Prismatic VLM vs. Deepseek VL).
>
> Our work undertakes a systematic study, combining different moving parts together. We hope to draw more robust and reliable conclusions and clarify the contradicting conclusions that exist in the MLLM domain. In the meantime, we aim to push the study of these modules to the extreme, especially in the fully open-source setting. For example, we carefully compare 15+ vision models, hoping to provide insights to both the MLLM and visual representation learning communities. We also collect and curate, to our knowledge, the largest open-source instruction tuning datasets. These efforts turn the findings in our work into pieces that narrow the gap between open-source and proprietary models.
>
> > **W3: Finding 7**
>
> A: Regarding Finding 7, we draw this conclusion based on multiple experiments rather than a single data point. We observed that, compared to ensembling only CLIP models, adding the DINOv2 model improves performance, especially on vision-centric benchmarks like RealWorldQA and CV-Bench. We appreciate the reviewer's feedback and will include more clarification on this finding in our revision.
>
>
> > **Q1: Question about model combination and two entries for "SigLIP+DINOv2+ConvNeXt"**
>
> A: Thank you for reviewing our work so carefully and catching this typo! The second instance of “SigLIP+DINOv2+ConvNeXt” uses a ConvNeXt-L not an XXL.
> We will correct this in the revised version of the draft.
>
> > **L1: Privacy of Internet Web Search**
>
> A: Thank you for raising this question! We absolutely value the importance of data privacy and copyright. We respect and approach this issue in the following two ways:
>
> *Collect Data from Licensed Websites*: Our web agent collects data from Wikipedia, which is licensed under CC BY-SA (https://en.wikipedia.org/wiki/Wikipedia:Copyrights). We attribute the data source in §5.1 and Appx. E.2.
> *Fully Open Source Data Collection*: We also fully open-source our data collection pipeline in Appx. E. This transparency allows for thorough inspection and verification, ensuring that our methods do not violate any copyright or data privacy regulations.
>
> Data privacy is a collective challenge faced by the community, and proprietary models often do not disclose details about their data pipeline. One of the aims of our project is to raise awareness and inspire new research on this topic by being *fully* transparent. We will release all details of the data collection pipeline and plan to include a section in the revision to discuss this further.

---

> ### Comment · Reviewer_K2Un · 2024-08-12
>
> Thank you for your clear responses. As most my concerns are addressed, I will maintain the original rating of accept.

---

### Official Review · Reviewer_xvQn · 2024-07-13

**Soundness:** 4
**Presentation:** 4
**Contribution:** 4
**Rating:** 8
**Confidence:** 4

**Summary:**

The paper conducts a comprehensive study of multimodal LLMs from a vision-centric perspective. Different from the lines of previous literature which aim to propose new architectures/algorithms for multimodal LLMs, this paper carefully splits the design space of visual parts of multimodal LLMs into several individual parts and diagnoses each with controlled experiments. This leads to several innovative conclusions about the visual aspects of multimodal LLMs, including the validity of standard benchmarks, the choices of visual encoders, etc.

**Strengths:**

Generally, the paper is a pleasure to read.

- The paper is well-motivated. MLLMs are differentiated from pure LLMs by the visual components. Thus it is quite natural to study the visual aspects of MLLMs.
- The paper draws rich connections from the visual representation learning literature, which I see as an original perspective, putting the paper into the appropriate position and delivering more valuable information to a broader community.
- The controlled analysis is precise and rigorous with carefully designed experiments. Particularly, the experiments start with an examination of existing benchmarks, which is a prerequisite of all following experiments.
- The conclusions are insightful and can be valuable for the future development of multimodal LLMs.

**Weaknesses:**

- The experiments only consider one particular LLaVa-like formulation of MLLMs built upon a pretrained LLMs, while Sota MLLMs like GPT-4o and Reka are more likely to have completely different training diagrams and architectures, e.g., treating images and texts equally and training a native multimodal LLMs from scratch, or training with interleaved visual and text contents instead of fixed image-first formulations. The value of the paper is thus limited.

-  A minor point: There is no analysis of why the findings hold.

**Questions:**

- What's your opinion of "native" multimodal LLMs? Do you think the findings in the paper will in a way transfer to more advanced models?

**Limitations:**

The authors have discussed the limitations quite adequately.

---

> ### Author Rebuttal · Authors · 2024-08-06
>
> We thank the reviewer for the thorough review and acknowledgments. We appreciate that you find our work “insightful and well-motivated”, contains “rigorous and carefully designed experiments”, and “can be valuable for the future development of multimodal LLMs”.
>
> > **W1 & Q1: Discussion around native MLLMs**
>
> A: We thank the reviewer for raising this question! We share our thoughts on these models below and show our findings in Cambrian are **very transferable** to these native models. We will add the discussion below to the revised version of our draft.
>
> **Thoughts about native MLLMs**
>
> We find native Multimodal LLMs that do not use a pre-trained vision encoder, like GPT-4o or Reka, to be an intriguing and promising approach. These native models have only recently been explored and are predominantly developed by proprietary companies such as OpenAI. As a result, their actual implementation, architecture, and training methods remain largely undisclosed. Additionally, there is insufficient evidence to assert that native MLLMs can overcome the limitations of current MLLMs, such as their visual deficiencies. On the other hand, vision-only representation learning itself is a significant and meaningful objective. Our study connects this goal with multimodal learning, providing scientific insights that complement future advancements in multimodal systems, whether they are native or not.
>
> We note that one major downside of native MLLMs is that they require *much* more data and computational resources, as they do not rely on the knowledge embedded in pretrained encoders.
>
> **Findings in Cambrian transfer to native MLLMs**
>
> We believe the findings and contributions in our work will continue to hold and guide the development of future models, including “native” MLLMs. Our findings regarding Data, Connectors, Evaluation, and Vision Backbones can be transferred in the following ways:
> - **Data**: The pool of instruction tuning data and our data curation studies can be very useful for supervised fine-tuning “native” MLLMs. Training native MLLMs is likely to still consist of pretraining, supervised fine-tuning, and RLHF. The data collection and curation insights can play an important role in both the pretraining and supervised fine-tuning stages.
> - **Connector**: The SVA module we proposed can be part of, or inspiration for, future native MLLM designs. The conflict between high-resolution features and a constrained number of tokens is likely to continue in native MLLMs. Therefore, the SVA module could be a competitive candidate for resolving this issue in such native MLLMs.
> - **Evaluation**: Our study on the “Multimodality” of benchmarks can help native MLLMs better assess their visual capability. The categorization of benchmarks also provides more organized and interpretable evaluation protocols for future works, especially in vision-centric domains.
> - **Vision Backbones**: Our study compares current visual representations, uncovering insights about training data (e.g., CLIP vs. SSL), training methods (Encoding vs. Generative), network architecture (ViT vs. ConvNext), and image resolutions. These insights can better guide developers when designing architecture, data, and methods for training native MLLMs.

---

> > ### Comment · Reviewer_xvQn · 2024-08-14
> >
> > Thanks for your reply! I appreciate your perspective on the native multimodal LLMs.

---

### Official Review · Reviewer_BwuE · 2024-07-30

**Soundness:** 3
**Presentation:** 4
**Contribution:** 3
**Rating:** 7
**Confidence:** 3

**Summary:**

The paper explores Multimodal Large Language Models (MLLMs) and constructs the Cambrian-1 series models. This approach builds a series of advanced MLLMs through five key pillars, achieving exceptional performance in vision-centric tasks and diverse benchmark tests. By exploring different visual encoders, the method designs a novel spatial-aware connector, SVA, to reduce the number of annotations and enhance the integration of visual and language models. The method also curates high-quality visual instruction fine-tuning data from public sources, emphasizing the importance of balanced data distribution and discussing various instruction fine-tuning strategies. Additionally, this paper critically analyzes and supplements existing MLLM benchmarks, introducing the vision-centric benchmark CV-Bench to more accurately evaluate the models' visual-centric capabilities. This approach achieves top performance across diverse benchmarks and excels in visual-centric tasks.

**Strengths:**

The paper aims to bridge the gap in visual understanding by exploring Multimodal Large Language Models (MLLMs) from a vision-centric perspective. By investigating various visual encoders, this method introduces an innovative spatial-aware connector, SVA, which minimizes the need for annotations and improves the synergy between visual and language models.  Additionally, the paper offers a critical assessment and enhancement of current MLLM benchmarks, presenting a new vision-centric benchmark, CV-Bench, to more precisely measure the models' visual-centric abilities. Cambrian-1 achieves top performance across diverse benchmarks and excels in visual-centric tasks. The paper is well written and the experiment is well solid.

**Weaknesses:**

No Weaknesses. See questions

**Questions:**

(1) As the number of cross-attention layers (D) and distinct groups of learnable queries (G) increases, the performance does not show continuous improvement (lines 256-258). It is worth exploring whether performance saturation occurs with the increase of (D) and (G).

(2) Instruction tuning data collected from open web maybe raise the potential data leakage. And provide a statistical analysis of the data categories.

(3) Table 2 suggests that increasing the value t does not continuously enhance performance. The proportion of data varies across different tasks. Additionally explore the data scaling law.

(4) From Cambrian 10M to 7M, whether higher data quality results in better model performance.

(5) Statistical analysis of the response length and the difficulty, diversity distribution of the instruction data.

(6) Compare the performence of current MLLMs like LLaVA and BLIP-2 using the same data and other models (i.e., Minicpm v2.5) with Cambrian-1.

(7) Evaluate model performance on high-resolution images or tasks with extreme aspect ratios, (i.e., V*Bench).

(8) Determine whether training the visual encoder in all tasks outperforms freezing it, and compare the convergence speed of end-to-end training versus two-stage training.

(9) Table 11 indicates that integrating more vision encoders does not necessarily lead to higher performance improvements, as seen with models like MMB, VStar, and MMEP.

(10) Provide detailed information of the parameter counts, training duration, and  training hyperparameters for different model backbones.

(11) The paper improves performance across various tasks by integrating most of the current vision encoders. Could a unified visual encoder be used instead?

(12) Some related work needs to be included and discussed.

Zhu D, Chen J, Shen X, et al. Minigpt-4: Enhancing vision-language understanding with advanced large language models[J]. arXiv preprint arXiv:2304.10592, 2023.

Wang W, Chen Z, Chen X, et al. Visionllm: Large language model is also an open-ended decoder for vision-centric tasks[J]. Advances in Neural Information Processing Systems, 2024, 36.

Chen G, Liu X, Wang G, et al. Tem-adapter: Adapting image-text pretraining for video question answer[C]//Proceedings of the IEEE/CVF International Conference on Computer Vision. 2023: 13945-13955.

Huang X, Wang J, et al. Segment and Caption Anything[C]//Proceedings of the IEEE/CVF Conference on Computer Vision and Pattern Recognition. 2024

Liu Y, Zhang C, et al. Universal Segmentation at Arbitrary Granularity with Language Instruction[C]//Proceedings of the IEEE/CVF Conference on Computer Vision and Pattern Recognition. 2024.

Gao P, Han J, Zhang R, et al. Llama-adapter v2: Parameter-efficient visual instruction model[J]. arXiv preprint arXiv:2304.15010, 2023.

**Limitations:**

The authors have discussed the limitations of this paper, and this paper has no direct negative societal impact.

---

> ### Author Rebuttal · Authors · 2024-08-07
>
> We thank the reviewer for the thorough review and acknowledgments. We appreciate that you find our work “bridges the gap in visual understanding”, “offers critical assessment and enhancement of MLLM benchmarks”, “introduces an innovative spatial-aware connector”, and “achieves top performance”. We summarize and respond to your questions below:
>
> > **Q1: Does performance saturate with the increase of (D) and (G) in SVA**
>
> A: We recognize the importance of this point and have conducted experiments to further investigate it. We tabulate results below. We observe performance improves with increasing D & G and saturates with D > 4 or G >3.
>
>  |D|G|OCR & Chart|
> |:-:|:-:|:-:|
> |2|1|52.1|
> |3|1|52.4|
> |4|1|52.8|
> |5|1|52.1|
> |3|1|52.4|
> |3|2|52.6|
> |3|3|53.1|
> |3|4|52.8|
>
> > **Q2: Data, Leakage, and effectiveness**
>
> A: Our data is collected from existing open-source works. Therefore, we prevent data leakage problems by carefully choosing only the training set of data sources. For our Internet Data Engine, we focus on only Wikipedia in our project, which does not overlap with our benchmarks.
> In the revision, we will add a section reporting the number of test images in each benchmark present in our final Cambrian-10M dataset by checking image hashes.
>
> > **Q3&4: Increasing data threshold t will not enhance the performance* and *whether higher data quality results in better performance**
>
> A: As we show in §5.2, data quality matters more than quantity. In Tab. 2, intermediate t values result in better performance. This result echoes observations in the data curation pipelines of CLIP and MetaCLIP
> Likewise, we observe that our higher-quality 7M subset results in better benchmark performance compared to training on all 10M data (Tab. 3). We believe this is a result of better balancing the dataset sources (Tab. 2, Fig. 14) and adjusting their relative sizes (Fig. 10).
>
> > **Q5: response length, difficulty, diversity of instruction-tuning data**
>
> A: Thank you for raising these questions! We have conducted further analysis on Cambrian-7M and -10M regarding data composition, length of questions, length of responses, and number of rounds in the instruction tuning data and summarized the results in Tab. 1 of the rebuttal material. As a result of data curation, the Cambrian-7M data are distributed similarly to the best data ratio we found in the data ratio experiment (Fig. 10).
>
> > **Q6: Compare current MLLMs using the same data**
>
> A: Like many previous studies in (M)LLMs, we argue that data is crucial in distinguishing different works. We conduct additional experiments with the LLaVA model trained using LLaMA-3-8b and Cambrian-7M data. Due to the short rebuttal period, we use the conventional 576 visual tokens of LLaVA-1.5, not the high-resolution approach of LLaVA-Next. Despite fewer tokens, this version performs comparably or better than LLaVA-Next in general, knowledge, and vision tasks. Adding our SVA further improves performance, especially in OCR & Chart tasks, while still using only 576 tokens.
>
> |Data|# Vis Token|General|Knowledge|OCR & Chart|Vision-Centric|
> |:--|--:|--:|--:|--:|--:|
> |LLaVA-Next|2880|72.5|55.6|61.0|55.0|
> |LLaVA w/ Cambrian Data|576|72.0|58.1|54.3|55.6|
> |Cambrian-8B|576|74.4|60.1|66.2|60.3|
>
> > **Q7: Evaluate on high resolution benchmarks such as V*Bench**
>
> A: In our work, we adopt V*Bench in our evaluation—see “V*Star” in Tabs. 4, 11, & 13. On V*Star, Cambrian-1 is competitive with LLaVA-NeXT and GPT-4V.
>
> > **Q8: Determine if unfreezing visual encoder outperforms freezing in all tasks, convergence speed**
>
> Unfreezing most visual encoders outperforms freezing in most tasks. We have added a visualization to the rebuttal material (Fig. 1) that shows the %change from Frozen to Unfrozen for each model on each benchmark. Note: full Frozen & Unfrozen benchmark results are in Tables 9 & 10.
>
> Thanks for raising the point about convergence speed. Given fixed compute, unfreezing is approximately 50-55% slower during fine-tuning. We will emphasize this drawback in the revision.
>
> > **Q9: Integrate more vision encoders does not lead to higher performance on every benchmark**
>
> A: Indeed, more vision encoders does not lead to higher performance on *every* benchmark. We believe this is expected, as different vision encoders have different strengths/weaknesses—as studied in §3.4 and Fig. 6—and thus different combinations of vision encoders inherit combinations of these strengths/weaknesses.
> We will amend the 7th Finding to clarify that combining multiple vision encoders usually enhances performance, *but not necessarily on every benchmark*.
>
> > **Q10: Provide more detail about parameter count and training hyperparameters**
>
> A: Tab. 14 provides training hyperparameters, including learning rate, batch size, weight decay, etc. We have added compute resources and training durations in the Rebuttal Tab. 2.
>
> > **Q11: Could a unified vision encoder be used**
>
> A: As studied in this work, we do not have a “perfect” encoder that excels in all areas (visual grounding, language understanding, high-resolution features, etc). Therefore, we pursue hybrid encoders, which can leverage the different strengths of several pretrained visual encoders. We acknowledge that hybrid vision encoders are a work-around solution to take advantage of various pretrained models—unified vision encoders could be trained from scratch with superior performance, but that would require much more compute and data.
> Overall, we advocate to use MLLMs as a vision model evaluator and hope to inspire the development of a unified and powerful vision model.
>
> > **Q12: More Related Works**
>
> A: We thank the reviewers for providing these new references. We will add these references to the revision. Specifically, we will add MiniGPT-4 and LLama-adapter V2 to our discussion of developing MLLMs, and we will add Visionllm, Tem-adapter, Segment and Caption Anything, and Universal Segmentation to our discussion on the downstream applications of MLLMs.

---

> > ### Comment · Reviewer_BwuE · 2024-08-13
> >
> > Thank you for the rebuttal. My concerns are addressed by the authors. Therefore, I will keep my score.

---

### Author Rebuttal · Authors · 2024-08-07

We thank all reviewers for their thorough review and valuable feedback on our paper. We appreciate that you find our work "bridges the gap in visual understanding" (Reviewers BwuE, xvQn), "offers assessment and enhancement of MLLM benchmarks" (Reviewers BwuE, K2Un,  ED7N), "well-written" (Reviewers K2Un, BwuE), “release a new dataset that benefits the community” (Reviewer ED7N), “proposes an innovative connector” (Reviewers BwuE, ED7N), “contains extensive and rigorous experiments” (Reviewers xvQn, ED7N), and "achieves top performance" (Reviewers BwuE, ED7N).

In the responses below, we address each reviewer’s questions individually. We encourage the reviewers to refer to the attached rebuttal PDF for a detailed review, including additional figures and experiment results encouraged by the reviews. We hope our responses address your questions. We look forward to engaging with you during the reviewer-author discussion period if you have any further questions.

---

### Decision · Program_Chairs · 2024-09-25

**Decision:**

Accept (oral)

**Comment:**

The paper presents Cambrian-1, a family of multimodal large language models with a vision-centric approach. The authors explore the use of various visual encoders and propose a novel spatial vision aggregator (SVA) to enhance the integration of vision and language models. The work critically evaluates existing benchmarks and curates a large dataset for visual instruction tuning. Cambrian-1 achieves state-of-the-art performance across multiple benchmarks and provides an open-source toolkit for further research in multimodal systems.

The contributions of the paper are solid. There are comprehensive evaluation of multimodal benchmarks, identifying gaps in visual understanding. The introduction of the SVA module, shows notable performance improvements in specific tasks like OCR and chart understanding. The open-source contributions including model weights, datasets, and code, will likely to benefit the research community considerably. Extensive experiments with various visual encoders offers valuable insights into their performance. Despite some limitations in novelty, the paper makes a valuable asset for the research community. The paper's performance on multiple tasks and its rigorous experimentation further support its acceptance.